# Financial crisis early warning of Chinese listed companies based on MD&A text-linguistic feature indicators

Zhishuo Zhang, Xinran Liu, Huayong Niu*

International Business School, Beijing Foreign Studies University, Beijing, China

* bfsuniuhy@163.com

**Data Availability Statement:** The data for this study can be downloaded at https://github.com/zzs0216/Financial-Crisis-Early-Warning-of-Chinese-Listed-Companies/tree/main (accessed on 19 August 2023).

## Abstract

Nowadays, the international situation is severe and complex, and the structural issues within the Chinese economy are prominent. Consequently, the financial risks faced by Chinese listed companies continue to escalate. Hence, it is of great practical significance to furnish effective early warnings for financial crises in listed companies. In this paper, we first employ web crawler technology and natural language processing technology to assess four text-linguistic features in the Management Discussion and Analysis (MD&A) section of the annual financial reports of listed companies in China from 2011 to 2020. These features are text tone, forward-looking, readability and similarity. Subsequently, we combine these features with traditional financial indicators and explore thirteen mainstream machine learning models to comparatively analyze their effectiveness in predicting financial crises in listed companies. The empirical findings of this research reveal that MD&A text readability and similarity indicators contribute valuable incremental information to prediction models, whereas text tone and forward-looking indicators exhibit the opposite effect. The latter two indicators can be manipulated more effortlessly by management, as this study's empirical findings indicate no evidence of their contributions in incremental informational value. In fact, the forward-looking indicator even introduces additional noise to the prediction. Finally, by comparing the early warning effects of thirteen machine learning models, it is found that RF, Bagging, CatBoost, GBDT, XGBoost and LightGBM models maintain stable and accurate sample recognition ability. In general, this paper constructs a more effective financial crisis early warning model by exploring the MD&A text-linguistic feature indicators, thereby offering a fresh research perspective for further investigations in this field.

## Introduction

Nowadays, the global economic recovery is sluggish, with a significant increase in regional conflicts and trade protectionism, which has further intensified global problems. Moreover, the Chinese economy is facing prominent structural issues. As a result, the complexity of the environment has increased, leading to a rise in uncertainty factors in the development process of Chinese listed companies, which makes the financial risks rise continuously. In such a

**Funding:** Funder: Zhishuo Zhang: Methodology, Writing—original draft preparation, Writing—review and editing. This work was supported by Beijing Foreign Studies University Double First Class Major Landmark Project [grant number 2022SYLZD001]; the Fundamental Research Funds for the Central Universities [grant number 2023JX095]; and Beijing Foreign Studies University G20 Research Center Project [grant number G20ZX20223003].

**Competing interests:** The authors have declared that no competing interests exist.

context, the construction of an effective financial crisis early warning model becomes crucial. Such a model will assist managers and operators of listed companies in proactively anticipate risks, conducting thorough risk analysis, implementing control measures, and taking timely countermeasures to resolve crises. Ultimately, this will improve the overall financial risk management level of Chinese listed companies and promote the healthy development of the capital market.

Financial crisis early warning generally starts with selecting some crisis identification indicators that are indicative of whether a future financial crisis will occur. Therefore, based on these indicators, statistical analysis or machine learning methods are used to predict it. In terms of crisis identification indicators selection, scholars primarily focus on structured financial indicators [1–5], while often ignoring unstructured textual indicators. However, as an important form of information disclosure, the text contains numerous linguistic feature information. Thus, the exploration of text-linguistic feature indicators can effectively supplement financial indicators and improve the overall accuracy and reliability of financial crisis early warning systems.

Management's Discussion and Analysis (MD&A) provides additional interpretation and analysis of the annual financial statements of a listed company, along with the management's explanation of the company's business plan for the upcoming year and their perspective on the opportunities and risks associated with future development [6–8]. In this paper, we assert that management, being insiders of a listed company, possess an information advantage, and the information they convey in the MD&A may contain incremental information. To explore incremental information in MD&A texts, we use natural language processing technology, and combine them with traditional financial indicators. Then thirteen mainstream machine learning models are applied to compare the early warning effects of listed companies' financial crises with different combinations of text-linguistic feature indicators as well as different machine learning models. Specifically, this paper aims to address the following six research questions: (1) whether the combination of MD&A text tone indicator and traditional financial indicators can improve the early warning effect on the financial crises of listed companies; (2) whether the combination of MD&A text forward-looking indicator and traditional financial indicators can improve the early warning effect on the financial crises of listed companies; (3) whether the combination of MD&A text readability indicator and traditional financial indicators can improve the early warning effect on the financial crises of listed companies; (4) whether the combination of MD&A text similarity indicator and traditional financial indicators can improve the early warning effect on the financial crises of listed companies; (5) whether the combination of MD&A text-linguistic feature indicators with enhanced warning effect and traditional financial indicators can improve the early warning effect on the financial crises of listed companies; (6) a comparative study on the early warning effectiveness of thirteen machine learning models.

Based on the above research questions, this paper uses a random undersampling method to select 832 listed companies of Chinese RMB common stocks (A shares) during 2011–2020 as the sample. And this paper takes the year of the sample as T and selects the traditional financial indicators in year T-3 as the benchmark. To be more specific, this study compares the effectiveness of financial crisis early warning of listed companies after incorporating MD&A text-linguistic feature indicators. Subsequently, this study compares the recognition performance of thirteen machine learning models for the samples. Finally, this study identifies text-linguistic feature indicators that can improve the recognition performance of listed companies' financial crises, along with the financial crisis early warning model with greater accuracy.

The main shortcomings of the existing studies are: (1) MD&A text tone and forward-looking information are susceptible to whitewashing, and there is controversy over whether text

tone and forward-looking information have incremental information value [9–14]; (2) at present, there is no study that uses the readability of MD&A texts for early warning of company financial crises; (3) there is no relevant research on the Chinese market using the MD&A text similarity indicator [15]; (4) existing studies on financial crisis early warning methods for listed companies primarily consist of comparative analyses utilizing a single model or a few models, with relatively few studies conducting comparisons using multiple machine learning models [16–19].

The main contributions of this paper are: (1) this paper assesses incremental information contained in various text-linguistic features of MD&A text and their impact on the financial crisis early warning for listed companies, using the same sample database. This analysis aims to address the limitations of existing research and offer a more comprehensive understanding of the enhancement effect provided by each text-linguistic feature indicator; (2) this study reveals that the text readability and similarity indicators among MD&A text-linguistic features contain incremental information, thereby enhancing the accuracy of early warning models for listed companies' financial crises; (3) by combining MD&A text-linguistic feature indicators with enhanced early warning effects, the research explores the effectiveness of financial crisis early warning in listed companies, thereby enriching the perspectives in this field; (4) thirteen mainstream machine learning models are compared, and six models are identified to exhibit superior and consistent early warning effects, offering valuable insights for further model users.

This paper is organized as follows. The second part reviews the relevant literature in this field. The third part presents an overview of the thirteen machine learning models employed in this study. The fourth part describes the selection process of traditional financial indicators and MD&A text-linguistic feature indicators, and also conducts an empirical study addressing six research questions. The fifth part offers further discussion of the empirical results. Lastly, the sixth part summarizes the key findings, acknowledges the limitations, and outlines future research directions.

## Related studies

### Financial crisis early warning indicators and methodological techniques

Financial crisis early warning typically involves the selection of predictors that signal the occurrence of a crisis will occur, followed by the application of statistical analysis methods or machine learning methods to detect such crises [20–24].

In terms of predictor selection, previous studies primarily focus on traditional financial indicators, mostly including five aspects: profitability, solvency, asset operating efficiency, cash flow quality and development quality [25–33]. A limited number of studies have identified the relationship between text-linguistic feature indicators and company performance [15, 34–36]. However, there is a dearth of literature that explores the combination of text-linguistic feature indicators with traditional financial indicators for early warning of financial crises in listed companies.

In terms of early warning methods for the financial crises of listed companies, relevant research has witnessed a progression from univariate analysis to multivariate analysis, and subsequently to the prevalent utilization of machine learning techniques. Fitzpatrick [37] employed univariate analysis and identified significant differences in the ratios of net income/shareholders' equity and shareholders' equity/debt three years prior to a company's bankruptcy. Beaver [38] developed a univariate model using a single ratio indicator for financial crisis early warning of a company. After that, Altman [39] applied multivariate discriminant analysis to financial crisis early warning, proposed the Z-score model, wherein the magnitude

of the Z value reflects the company's bankruptcy risk. Then Altman et al. [40] further enhanced the Z-score model with the introduction of the ZETA model, which demonstrated significant improvements in long-term financial crisis early warning effectiveness. Ohlson [41] employed a Logistic regression model, overcoming the stringent requirements for independent variable distribution and gradually replacing discriminant analysis as the mainstream method in financial crisis early warning. With the development of information technology, machine learning methods were incorporated into financial crisis early warning. Odom and Sharda [16] utilized a neural network to construct a financial crisis early warning model, which exhibited enhanced predictive capabilities. Scholars explored the application of the random forest model for financial crisis early warning of listed companies, and observed significant improvements compared to the adaptive boosting algorithm [17]. Comparative analyses of the adaptive boosting algorithm and neural network model for financial crisis early warning, using European companies as samples, revealed a reduction of approximately 30% in generalization error with the adaptive boosting algorithm [18]. Wang et al. [19] employed three decision tree models to construct a financial crisis early warning model and found that the random forest had superior classification results.

In summary, the majority of existing research methods focus on single models or a comparative study involving a limited number of models, with relatively limited exploration of comparative studies using multiple machine learning models.

## Text-linguistic feature indicators and financial crisis early warning

The development of information technology has led to increasingly comprehensive mechanisms and content for company information disclosure, prompting a gradual exploration of text-linguistic feature indicators.

Text tone, as a important non-content textual information, partially reflects management's attitude and sentiment towards the company's business and future prospects. However, the incremental information value of text tone remains a subject of controversy [42, 43]. On one hand, certain studies argue that text tone is informative and can predict a company's future earnings [44, 45], performance [46–49], and even aid in financial crisis identification [50–52]. On the other hand, it has been suggested that management may manipulate text tone to whitewash negative information, such as engaging in earnings management activities [9], thereby increasing operational risks [10] and diminishing the accuracy of early warning systems for financial crises through text tone [11]. Consequently, the existing literature has not definitively established whether text tone possesses incremental information value for identifying financial crises.

Forward-looking information, characterized by its prospective and textual nature, is more intricate and ambiguous compared to historical and quantitative information [53]. The incremental information value of text forward-looking information also remains a topic of debate. Several studies have affirmed its effectiveness, suggesting that it provides incremental information value, predicts future performance [44], and offers risk warnings [54]. However, the immediate verification and auditing of forward-looking information are challenging [12], making it less reliable. Some scholars argue that forward-looking information is susceptible to "linguistic inflation" to varying degress, diminishing its value [13, 14]. Consequently, the incremental information value of text forward-looking information is currently an unresolved matter that requires urgent researches, and its role in early warning system for financial crises remains controversial.

Text readability, as a fundamental characteristic of textual information, reflects the comprehensibility of the information for text users. The readability of texts disclosed by companies

directly affects the efficiency of investors' information access [55]. During periods of underperformance, management has greater incentives to obfuscate unfavorable information by employing strategic disclosures and unconventional language to obscure statements [56, 57]. Lo et al. [58] found that management tends to reduce text readability to conceal earnings management activities, while Li [36] observed that companies with poor earnings persistence produce less readable annual reports. It is reasonable to assume that companies disclosing less readable information are more likely to suffer financial crises. However, there have been no studies thus far that utilize text readability to predict company financial crises.

Text similarity, unlike text tone and forward-looking information that can be intentionally overlooked, directly reflects the extent to which a company information deviates from historical information. Text similarity often indicates the company management's disclosure willingness, capability, or operational aspects. Firstly, motivated by the desire to conceal negative news, company management may adopt a passive stance towards information disclosure, prioritizing form over substance [59]. Secondly, higher text similarity may stem from low transparency within the company, hindering effective information transfer, or management's lack of sufficient and effective analytical skills [60]. Thirdly, high similar information may suggest that the company's operating conditions in the current or future periods show little significant changes compared to the past [59]. Cohen et al. [15], in their study of the U.S. market, found that changes in the language and structure of financial reports have a strong impact on future returns and operations of companies, with the degree of these changes predicting future earnings, profitability, financial crises and even company bankruptcies. However, no relevant studies have been conducted for the Chinese market.

Based on the literature review, it is observed that text readability and similarity generally provide incremental information value, with low readability and high similarity indicating poor company performance, while the interpretation of text tone and forward-looking information remains a subject of controversy.

## Empirical analysis

### Design of the empirical analysis process

The empirical process of financial crisis early warning of Chinese listed companies based on MD&A text-linguistic feature indicators is shown in Fig 1.

In the first step, this paper performs index design, which involves selecting traditional financial indicators and constructing MD&A text-linguistic feature indicators. The traditional financial indicators form the first input feature, while the other four features consist of four types of MD&A text-linguistic feature indicators and traditional financial indicators.

In the second step, thirteen machine learning models are used to sequentially substitute the five input feature variables derived in the first step into the model. The output feature variables are binary values, i.e., whether the companies are listed as ST-listed.

In the third step, the effectiveness of different input feature variables in identifying financial crises of listed companies is individually compared against the benchmark of the first input feature.

In the fourth step, this paper identifies the combination of MD&A text-linguistic feature indicators that enhance early warning effects. These indicators are then combined with traditional financial indicators to further evaluate their effectiveness in identifying financial crises of listed companies.

In the fifth step, the recognition performance of thirteen machine learning models for samples is compared based on different input feature variables.

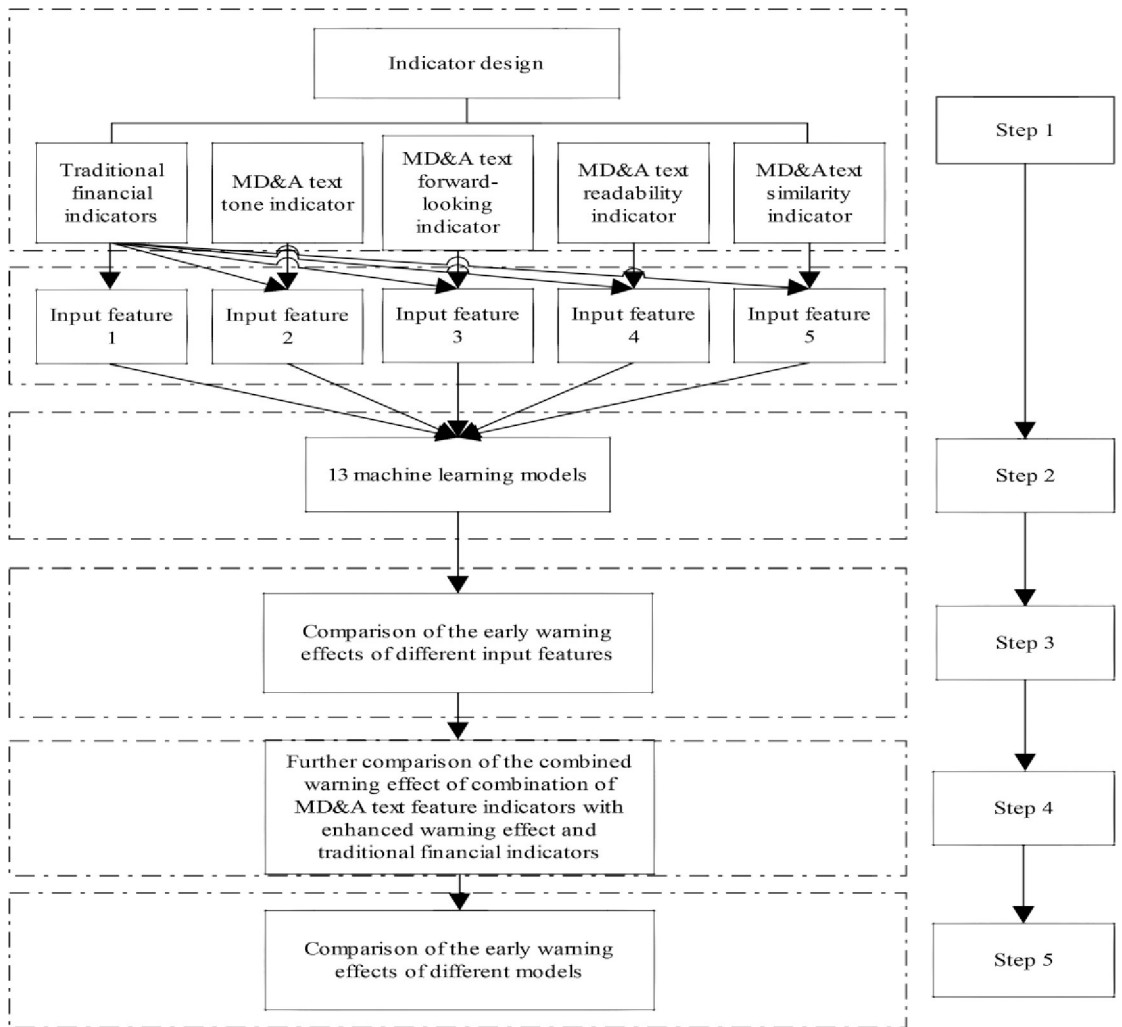

**Fig 1. Flow chart of empirical analysis.**

### Evaluation index system construction

In this chapter, the selection of traditional financial indicators and the construction of MD&A text-linguistic feature indicators are explained.

**Traditional financial indicators.** Given the absence of an established standard for financial indicators in financial crisis early warning models, this study references the findings of previous scholars [25–33]. Five financial aspects, namely profitability, solvency, asset operating efficiency, cash flow quality and development quality, are chosen, encompassing a total of 10 specific indicators. The selection and calculation methods of traditional financial indicators are presented in Table 1.

The selection of these ten indicators as financial indicators for assessing the financial crisis early warning of listed companies is based on the following considerations: (1) Earnings per share: earnings per share refers to the indicator obtained by dividing a company's net profit by the weighted average number of shares issued. It is an important indicator for measuring the profitability associated with each share of a company's stock. A higher Earnings per share indicates a strong profitability of the company, while a lower Earnings per share may suggest a

**Table 1. The selection of traditional financial indicators and formulas.**

| No. | Traditional Financial Indicators | Formulas |
|-----|----------------------------------|----------|
| 1 | Earnings per share (yuan) | The company shall calculate basic earnings per share by dividing the net profit for the period attributable to common shareholders by the weighted average number of common shares outstanding |
| 2 | Operating profit ratio (%) | Operating profit/Operating income × 100% |
| 3 | Debt assets ratio (%) | Total liabilities/Total assets × 100% |
| 4 | Current ratio (time) | Current assets/Current debts |
| 5 | Cash coverage ratio (%) | Net cash flow from operating activities/Current debts × 100% |
| 6 | Interest cover (time) | (Net income + Income tax expense + Finance costs) Finance costs |
| 7 | Accounts receivable turnover ratio (time) | Operating revenues/Accounts receivable ending balance |
| 8 | Sales-to-cash ratio (time) | Cash received from sales of goods and services/Operating income |
| 9 | Operating cash per share (yuan) | Net cash flow from operating activities/Total common share capital at the end of the year |
| 10 | Net profit growth rate (time) | Additional net profit in this year/Net profit in the previous year |

decline in profitability and the presence of financial risks. (2) Operating profit ratio: operating profit ratio refers to the ratio of operating profit to operating revenue, reflecting the company's ability to generate profits in its operations. A high operating profit ratio indicates strong profitability, while a low operating profit margin may suggest a decline in profitability and potential financial crisis. (3) Debt assets ratio: debt assets ratio represents the proportion of a company's assets that are financed through debt. It can reflect the company's debt risk and financial stability. A higher debt assets ratio indicates a higher level of debt risk for the company, while a lower debt assets ratio generally indicates higher financial stability for the company. (4) Current ratio: current ratio is a measure of a company's short-term debt-paying ability and liquidity risk. It assesses the company's ability to meet short-term liabilities by comparing its current assets to its current liabilities. A higher current ratio indicates that the company has sufficient current assets to cover short-term debts, while a lower current ratio may suggest insufficient liquidity and potential difficulties in meeting debt obligations, which can lead to a financial crisis. (5) Cash coverage ratio: cash coverage ratio measures the relationship between the net cash flow generated from operating activities and the ability to repay current liabilities in a listed company. It reflects the extent to which the company's cash flow covers short-term debt obligations. A higher cash coverage ratio indicates a stronger ability to repay debts using cash flow, while a lower ratio may suggest insufficient cash flow, making it difficult for the company to repay debts on time, potentially leading to a financial crisis. (6) Interest cover: interest cover is a measure of a company's ability to pay interest expenses. It represents the profitability that a company can pay interest expense before interest is paid. A higher interest cover indicates a stronger ability to cover interest payments, while a lower interest cover may suggest difficulties in paying interest, indicating that shareholder equity could be affected and indicating the possibility of a financial crisis. (7) Accounts receivable turnover ratio: accounts receivable turnover ratio reflects how quickly a company collects its accounts receivable and relates to its cash flow recovery capability as well as the risk associated with accounts receivable. A higher ratio indicates that the company is able to collect accounts receivable quickly and utilize funds efficiently. On the other hand, a lower ratio may suggest poor efficiency in collecting accounts receivable, indicating potential financial strain and bad debt risk. (8) Sales-to-cash ratio: sales-to-cash ratio reflects the proportion of cash received from the sale of goods or services to the total operating revenue of a company. A higher ratio indicates that the company can quickly

convert sales revenue into cash income, while a lower ratio may suggest the company faces risks of delayed payments or insufficient liquidity. (9) Operating cash per share: operating cash per share represents the relationship between the cash flow generated from a company's operating activities and the total common shares outstanding at the end of the year. It measures the stability and sustainability of a company's operating cash flow and provides investors with more insightful financial information. (10) Net profit growth rate: net profit growth rate reflects changes in a company's profitability and reveals the trend of its profit-generating ability. A higher net profit growth rate has a positive impact on the financial stability and sustainability of a company, while a lower rate may indicate a decline in profitability and potential financial challenges.

In summary, these ten traditional financial indicators comprehensively consider profitability, solvency, asset operating efficiency, cash flow quality, and development quality of listed companies. They have a high level of reliability and significance in predicting financial crises, aiding in the assessment a company's financial condition and early warning of potential risks.

**MD&A text-linguistic feature indicators.** In this study, a Python web crawler program is implemented to crawl MD&A text from the Cninfo website (http://www.cninfo.com.cn/new/index). The obtained MD&A text is then processed through data cleaning, Chinese word segmentation, and other step to convert it into plain text. Subsequently, text-linguistic feature indicator is extracted from the cleaned MD&A text.

*(1) MD&A text tone indicator.* In this paper, we utilize the financial emotion English words list provided by Loughran & McDonald [46]. We translate the English words in this list to create a Chinese emotion dictionary and perform statistical analysis on positive negative emotion words.

In terms of text tome measurement, existing literature presents two primary metrics. One approach measures text tone by calculating the ratio of the difference between positive and negative tone words to the total vocabulary [61, 62]. The other approach calculates the ratio of the difference between positive and negative words to the sum of positive and negative words [63–68]. The latter method is widely used in current literature. Therefore, this study adopts the second method to measure text tone, as depicted in Eq (1).

where *Tone* is the tone value, which takes values in the range [−1,1]. *Pos* is the number of positive tone words in the text. *Neg* is the number of negative tone words in the text.

$$Tone = \frac{Pos - Neg}{Pos + Neg} \tag{1}$$

*(2) MD&A text forward-looking indicator.* The MD&A text forward-looking indicator pertains to the information contained in the text that is relevant to the future development and performance of the company. In this study, we employ the forward-looking keyword set proposed by Muslu et al. [69] and Zhang et al. [70]. The seed word set is expanded using the Word2Vec machine learning technique to include similar words. Subsequently, the word frequencies of the forward-looking keywords are calculated, and the forward-looking indicator is derived by dividing the number of word frequencies by the total number of words in the text.

*(3) MD&A text readability indicator.* The text readability indicator influences how quickly and thoroughly users understand textual information. In this study, we assess this indicator using the utterance production probability method proposed by Shin et al. [71]. This method primarily relies on the Word2vec neural network model developed by Mikolov

et al. [72]. In this paper, the production probability of the current word $P_w$ is calculated based on the semantic features and contextual position of the words in the text. The production probability of the whole utterance $P_s(P_w)$ is obtained by concatenating and multiplying the production probability of each word. $N$ represents the number of sentences constituting the text. The specific calculation formula is given in Eq (2).

$$Readability = \frac{1}{N}\sum_{s=1}^{N} \log P_s(P_w) \tag{2}$$

*Readability* is the value of readability. The higher this value, the higher the average probability of sentence generation in the whole text, and the easier the text is to be understood, i.e., the higher the readability of the text.

*(4) MD&A text similarity indicator*. The MD&A text similarity indicator measures the degree of semantic approximation in content between the current and the previous MD&A text information. In this study, we employ the word frequency-inverse document frequency method [73] to determine the weights assigned to each word item in both the current and previous text issues. Assuming that $v_a$ is the keyword term vector of the current issue MD&A text and $v_b$ is the keyword term vector of the previous issue MD&A text, with $Similarity_{AB} = \cos(\alpha) = \frac{V_1}{V_1} * \frac{V_2}{V_2} = \frac{V_1 * V_2}{V_1 V_2}$ and $v_b = (w_{b1}, w_{b2}, \cdots, w_{bn})$, the formula for calculating the similarity between the two issues of MD&A texts is shown in Eq (3).

$$Similarity = \mathrm{Cos}(\alpha) = \frac{V_a}{\|V_a\|} * \frac{V_b}{\|V_b\|} = \frac{V_a * V_b}{\|V_a\|\|V_b\|} \tag{3}$$

## Sample selection and source of data

Regarding financial crisis sample selection, scholars commonly consider a listed company being specially treated (ST) as a primary sign of being in a financial crisis [1–4]. Effective financial crisis warnings should have an early detection capability. Compared to the failure to meet debt obligations and bankruptcy, listed companies are often labeled as ST at an earlier stage. Thus, identifying ST status is more effective as an early warning. Therefore, this study selects companies in Shanghai and Shenzhen that firstly labeled as ST-listed from 2011 to 2020 as the sample for financial crises, totaling 481 companies over a 10-year period. The year in which a listed company experiences a financial crisis is defined as year T. ST refers to listed companies that have received special treatment for two consecutive years of losses, so the data in year T-2 can already show a financial crisis. To avoid overestimating the model's prediction ability, this study utilizes data in year T-3 instead of year T-2, i.e., 2008–2017 data of both traditional financial indicators and text-linguistic feature. The selection of 2008 as the starting point is primarily driven by the implementation of new accounting standards for Chinese listed companies in 2007. This brought significant changes in the meaning and calculation methods of traditional financial indicators, resulting in lower comparability between data before and after 2007. Simultaneously, the global financial crisis erupted in 2008, leading to substantial changes in both the Chinese domestic and international economic environment. In response, China proactively provided new interpretations and adjustments to annual reports under the new accounting standards. This action enhanced the comparability of data after 2008. Additionally, the disclosure standards for MD&A texts were standardized after 2008, further contributing to the improved comparability of data after 2008. Therefore, based on these factors, we have chosen 2008 as the starting point for our research.

Subsequently, this paper excludes samples of listed companies in finance and insurance, as well as those with missing data on traditional financial indicators in year T-3 and missing text similarity. This results in a final sample size of 416 ST-listed companies. Considering the unbalanced number of ST-listed companies and non-ST-listed companies, this paper will explain the treatment in section Data Processing and Descriptive Statistics. The total number of processed samples is 832, comprising 416 non-ST-listed companies and 416 ST-listed companies. The traditional financial indicators data are obtained from the China Economic and Financial Research Database, while the MD&A text is crawled from the Cninfo website and processed into MD&A text-linguistic feature indicators through natural language processing technology.

## Data processing and descriptive statistics

To address the sample imbalance issue, this study employs the random undersampling method, a widely applied approach by scholars in similar cases [74–76], i.e., randomly removing some financially normal listed companies from the sample so that the number of ST-listed companies is the same as the number of non-ST-listed companies. This forms a balanced dataset to be used in the subsequent training phase of the machine learning model.

When utilizing ridge regression and lasso regression, the presence of regularization terms in the models leads to smaller coefficients for feature variables with larger magnitudes. Consequently, these variables may be overlooked. To mitigate this effect, normalization is applied to all input feature variables. In addition, since gradient descent is used for loss function optimization, normalization aids in achieving faster convergence and a more straight, stable gradient descent process. So we normalize all input feature variables according to Eq (4).

$$x_i^* = \frac{x_i - x_{\min}}{x_{\max} - x_{\min}} \tag{4}$$

$x_i^*$ denotes the value of the feature variable data after normalization, and the range is [0,1]. $x_i$ denotes the value of the feature variable data before normalization. $x_{\min}$ and $x_{\max}$ denote the minimum and maximum values in the feature variable data, respectively.

The descriptive statistics of traditional financial indicators and text-linguistic feature indicators are shown in Table 2.

## Machine learning model early warning effect evaluation indicator

In this paper, we discuss the binary classification problem within supervised learning. It involves determining the true type of a sample and comparing it to the type predicted by the model, resulting in four possible scenarios: positive samples predicted by the model as True Positive (*TP*), negative samples predicted by the model as False Positive (*FP*), positive samples predicted by the model as False Negative (*FN*), and negative samples predicted by the model as True Negative (*TN*). In this study, ST-listed companies are considered positive samples, while non-ST-listed companies serve as negative samples. Based on four scenarios, we further calculate the accuracy, sensitivity, specificity, and AUC value as the evaluation indexes to assess the early warning effect of the machine learning model.

*Accuracy* measures the proportion of positive samples and negative samples that are correctly predicted to the total sample and is calculated by the formula in Eq (5).

$$Accuracy = \frac{TP + TN}{TP + FP + FN + TN} \tag{5}$$

*Sensitivity* measures the accuracy of the model in predicting ST-listed companies. The

**Table 2. Descriptive statistics of traditional financial indicators and text-linguistic feature indicators.**

| Variable | Obs | Mean | Std. Dev. | Min | Max |
|---|---|---|---|---|---|
| Earnings per share | 832 | 0.14 | 0.57 | -5.88 | 4.49 |
| Operating profit ratio | 832 | -0.21 | 4.18 | -116.76 | 13.97 |
| Debt assets ratio | 832 | 0.56 | 0.43 | 0.02 | 8.00 |
| Current ratio | 832 | 1.45 | 1.57 | 0.00 | 21.12 |
| Cash coverage ratio | 832 | 0.09 | 0.52 | -9.22 | 4.31 |
| Interest cover | 832 | 48.67 | 1,244.86 | -1,471.08 | 35,725.11 |
| Accounts receivable turnover ratio | 832 | 329.61 | 4,671.90 | 0.27 | 124,214.10 |
| Sales-to-cash ratio | 832 | 1.02 | 0.90 | 0.12 | 25.15 |
| Operating cash per share | 832 | 0.22 | 0.85 | -6.65 | 9.42 |
| Net profit growth rate | 832 | -131.71 | 1,301.70 | -23,572.69 | 13,737.70 |
| Tone Indicator | 832 | 0.40 | 0.17 | -0.22 | 1.00 |
| Forward-Looking Indicator | 832 | 0.01 | 0.00 | 0.00 | 0.02 |
| Readability Indicator | 832 | -21.88 | 4.26 | -42.82 | -11.20 |
| Similarity Indicator | 832 | 0.60 | 0.16 | 0.03 | 0.97 |
| ST (Not St is 1, St is 2) | 832 | 1.50 | 0.50 | 1 | 2 |

formula is shown in Eq (6).

$$Sensitivity = \frac{TP}{TP + FN} \qquad (6)$$

*Specificity* measures the accuracy of the model in predicting non-ST-listed companies. The formula is shown in Eq (7).

$$Specificity = \frac{TN}{TN + FP} \qquad (7)$$

*AUC* value represents the area under the Receiver Operating Characteristic (ROC) Curve, which evaluates the classifier's capability to distinguish between positive and negative samples. It is a performance indicator of the classifier's merit. A higher *AUC* value indicates a better classification performance of the model is. The calculation formula for the *AUC* value is displayed in Eq (8).

$$AUC = \frac{\sum_{i \in positiveClass} rank_i - \frac{m(1+m)}{2}}{m*n} \qquad (8)$$

Where $rank_i$ denotes the ordinal number of the $i$-th sample. $m$ and $n$ denote the number of positive and negative samples, respectively. $\sum_{i \in positiveClass}$ denotes that only the ordinal numbers of the positive samples are added up.

## Results of empirical analysis

To enhance the generalization ability of the machine learning model and prevent overfitting, a 10-fold cross-validation method is employed in this study. This method divides the sample dataset into 10 copies, and takes turns to using 9 of them as the training set and 1 as the test set. The training set is used to train the machine learning model, while the test set evaluates its performance. To further increase the reliability and reduce result variability, a 10 times 10-fold cross-validation is conducted, and the averaged results serve as the estimation of the model's effect.

In this chapter, we initially utilize traditional financial indicators as input feature variables to predict financial crisis. Subsequently, text-linguistic features, namely MD&A text tone, forward-looking, readability and similarity indicators, are added respectively to assess their potential enhancements on prediction accuracy. After that, by combining traditional financial indicators with MD&A text-linguistic feature indicators that were identified as having enhancement effects in the preceding analysis, we further examine the enhancement effects of these indicators being identified as enhanced indicators on early warning capabilities. Finally, we conduct a comparative analysis of the sample recognition performance of thirteen machine learning models to evaluate the models with better predictive performance. This experiment is conducted using Python 3.8.

**Analysis of empirical results of financial crisis early warning based on traditional financial indicators.** This paper first conducts a test on the early warning effect of thirteen machine learning models using traditional financial indicators. The detailed results are shown in Table 3, reflecting the accuracy, specificity, sensitivity, and AUC of thirteen machine learning models. The last column of the table ranks the models based on the magnitude of their AUC values (the same below). As can be seen in the table, the top five models with the highest AUC values are RF, Bagging, CatBoost, XGBoost and GBDT. Among them, RF achieves the highest prediction accuracy of 0.6394 for the sample of ST-listed companies, followed by XGBoost and GBDT, both at 0.6346. Among the thirteen machine learning models, DT exhibits the lowest AUC value of 0.6588 and the lowest sensitivity of 0.5313, indicating its limited effectiveness in identifying financial crises of listed companies. All thirteen machine learning models have AUC values of 0.65 or higher, with the mean of 0.6904. The mean accuracy, specificity and sensitivity are 0.6313, 0.6505, and 0.6121, respectively.

**Analysis of empirical results of financial crisis early warning based on traditional financial indicators and MD&A text tone indicator.** This section examines whether the MD&A text tone indicator and traditional financial indicators can improve the early warning effect for listed companies' financial crises. The two types of indicators are used as input feature variables, and their early warning effects are compared with those of the model using only traditional financial indicators. The results, are shown in Table 4, indicate that the differences in the mean values of accuracy, specificity, sensitivity, and AUC values measured by thirteen machine learning models are not statistically significant. The increase in accuracy is 0.0152,

**Table 3. Early warning effect based on traditional financial indicators.**

|  | Accuracy | Specificity | Sensitivity | AUC | Rank |
|---|---|---|---|---|---|
| Logistic regression | 0.6478 | 0.6635 | 0.6322 | 0.6910 | 6 |
| Ridge regression | 0.6250 | 0.6466 | 0.6034 | 0.6775 | 12 |
| Lasso regression | 0.6334 | 0.6466 | 0.6202 | 0.6810 | 10 |
| SVM | 0.6322 | 0.6563 | 0.6082 | 0.6785 | 11 |
| BPNN | 0.6202 | 0.6322 | 0.6082 | 0.6744 | 13 |
| DT | 0.6022 | 0.6731 | 0.5313 | 0.6588 | 14 |
| GBDT | 0.6370 | 0.6394 | 0.6346 | 0.6987 | 5 |
| CatBoost | 0.6334 | 0.6538 | 0.6130 | 0.7101 | 3 |
| XGBoost | 0.6358 | 0.6370 | 0.6346 | 0.7011 | 4 |
| LightGBM | 0.6202 | 0.6394 | 0.6010 | 0.6895 | 8 |
| AdaBoost | 0.6334 | 0.6611 | 0.6058 | 0.6880 | 9 |
| RF | 0.6466 | 0.6538 | 0.6394 | 0.7154 | 1 |
| Bagging | 0.6394 | 0.6538 | 0.6250 | 0.7107 | 2 |
| **Mean** | 0.6313 | 0.6505 | 0.6121 | 0.6904 | — |

Table 4. Early warning effect based on traditional financial indicators and MD&A text tone indicator.

| | Accuracy | Specificity | Sensitivity | AUC | Rank |
|---|---|---|---|---|---|
| Logistic regression | 0.6563 | 0.6731 | 0.6394 | 0.6984 | 8 |
| Ridge regression | 0.6358 | 0.6538 | 0.6178 | 0.6827 | 10 |
| Lasso regression | 0.6502 | 0.6659 | 0.6346 | 0.6891 | 9 |
| SVM | 0.6322 | 0.6394 | 0.6250 | 0.6766 | 12 |
| BPNN | 0.6394 | 0.6202 | 0.6587 | 0.6812 | 11 |
| DT | 0.6250 | 0.6827 | 0.5673 | 0.6641 | 13 |
| GBDT | 0.6575 | 0.6466 | 0.6683 | 0.7078 | 4 |
| CatBoost | 0.6550 | 0.6755 | 0.6346 | 0.7140 | 3 |
| XGBoost | 0.6466 | 0.6514 | 0.6418 | 0.7036 | 5 |
| LightGBM | 0.6394 | 0.6563 | 0.6226 | 0.6990 | 7 |
| AdaBoost | 0.6502 | 0.6563 | 0.6442 | 0.6992 | 6 |
| RF | 0.6671 | 0.6683 | 0.6659 | 0.7220 | 1 |
| Bagging | 0.6490 | 0.6587 | 0.6394 | 0.7164 | 2 |
| **Mean** | 0.6464 | 0.6575 | 0.6354 | 0.6965 | — |

specificity is 0.0070, sensitivity is 0.0233 and AUC is 0.0061. Although there is a slight increase in the mean values, some models show a decrease in certain evaluation indicators.

The lack of evidence indicates that text tone can enhance the effect of financial crisis early warning, thereby undermining the view that text tone contains incremental information. We contend that the presence of management manipulation and modification of text tone in ST-listed companies poses challenges in using text tone as a discriminative variable between ST and non-ST-listed companies. Based on the empirical results, we can clearly see that text tone indicator does not exert a significant effect on the financial crisis early warning of listed companies.

**Analysis of empirical results of financial crisis early warning based on traditional financial indicators and MD&A text forward-looking indicator.** We examine whether MD&A text forward-looking indicator can improve the effectiveness of traditional financial indicators in identifying of companies' financial crises in this section. To do so, we substitute the input feature variables with traditional financial indicators and MD&A text forward-looking indicators. The early warning effect is analyzed using the same thirteen machine learning models. The specific results are shown in Table 5. By comparing to the early warning effect of thirteen machine learning models using traditional financial indicators, we observe a decrease in the average values of accuracy, specificity, and AUC by 0.0026, 0.0080 and 0.0043, respectively. Although there is a slight increase in the overall mean of sensitivity, rising from 0.6121 to 0.6148, the increment amounts to only 0.0027. Specifically, all thirteen machine learning models exhibit a decrease in AUC. Besides, the accuracy of nine machine models, specificity of eleven models, and sensitivity of five models have declined.

Based on the empirical findings, it is evident that the MD&A text forward-looking indicator does not have a significant impact on the early warning of financial crises in listed companies. The author posits that the presence of strategic statements, such as the deliberate disclosure of more forward-looking information by ST-listed companies, complicates the utilization of text forward-looking indicators as differentiating variables between ST and non-ST-listed companies. Consequently, forward-looking text is deemed redundant information rather than incremental information.

**Analysis of empirical results of financial crisis early warning based on traditional financial indicators and MD&A text readability indicator.** Rutherford [56] found that

**Table 5. Early warning effect based on traditional financial indicators and MD&A text forward-looking indicator.**

|  | Accuracy | Specificity | Sensitivity | AUC | Rank |
|---|---|---|---|---|---|
| Logistic regression | 0.6394 | 0.6514 | 0.6274 | 0.6868 | 8 |
| Ridge regression | 0.6214 | 0.6346 | 0.6082 | 0.6723 | 11 |
| Lasso regression | 0.6298 | 0.6298 | 0.6298 | 0.6774 | 9 |
| SVM | 0.6094 | 0.5962 | 0.6226 | 0.6675 | 12 |
| BPNN | 0.6226 | 0.6563 | 0.5889 | 0.6734 | 10 |
| DT | 0.6034 | 0.6635 | 0.5433 | 0.6549 | 13 |
| GBDT | 0.6274 | 0.6370 | 0.6178 | 0.6926 | 4 |
| CatBoost | 0.6514 | 0.6635 | 0.6394 | 0.7045 | 3 |
| XGBoost | 0.6274 | 0.6322 | 0.6226 | 0.6923 | 5 |
| LightGBM | 0.6202 | 0.6274 | 0.6130 | 0.6869 | 7 |
| AdaBoost | 0.6310 | 0.6538 | 0.6082 | 0.6875 | 6 |
| RF | 0.6394 | 0.6442 | 0.6346 | 0.7154 | 1 |
| Bagging | 0.6502 | 0.6635 | 0.6370 | 0.7075 | 2 |
| **Mean** | 0.6287 | 0.6426 | 0.6148 | 0.6861 | — |

management improves the readability of disclosures during favorable company performance. However, in times of poor performance, complex and unconventional vocabularies are used to obscure the presentation. Thus, the readability of MD&A text may strengthen the efficacy of traditional financial indicators for detecting financial crises in listed companies. The results are shown in in Table 6. By comparing to the warning effects of thirteen machine learning using traditional financial indicators, all thirteen machine learning models exhibit positive changes in accuracy, specificity, sensitivity, and AUC with MD&A text readability indicator added. The overall average value experiences a substantial increase, specifically 0.0867, 0.1259, 0.0475 and 0.0897, respectively. In addition, in the original dataset, the mean value of the text readability indicator is -23.1005 for ST-listed companies and -20.6662 for non-ST-listed companies. This indicates a low MD&A text readability for ST-listed companies.

The empirical results demonstrate the significant impact of the MD&A text readability indicator on financial crisis identification. Specifically, lower readability in company disclosures

**Table 6. Early warning effect based on traditional financial indicators and MD&A text readability indicator.**

|  | Accuracy | Specificity | Sensitivity | AUC | Rank |
|---|---|---|---|---|---|
| Logistic regression | 0.7007 | 0.7596 | 0.6418 | 0.7402 | 10 |
| Ridge regression | 0.6923 | 0.7428 | 0.6418 | 0.7312 | 13 |
| Lasso regression | 0.6899 | 0.7404 | 0.6394 | 0.7341 | 11 |
| SVM | 0.6935 | 0.7428 | 0.6442 | 0.7320 | 12 |
| BPNN | 0.7212 | 0.8293 | 0.6130 | 0.7649 | 9 |
| DT | 0.6995 | 0.7524 | 0.6466 | 0.7782 | 8 |
| GBDT | 0.7404 | 0.8077 | 0.6731 | 0.8098 | 4 |
| CatBoost | 0.7524 | 0.8462 | 0.6587 | 0.8208 | 2 |
| XGBoost | 0.7296 | 0.7692 | 0.6899 | 0.8095 | 5 |
| LightGBM | 0.7127 | 0.7644 | 0.6611 | 0.8054 | 6 |
| AdaBoost | 0.7103 | 0.7428 | 0.6779 | 0.7806 | 7 |
| RF | 0.7524 | 0.8173 | 0.6875 | 0.8217 | 1 |
| Bagging | 0.7392 | 0.7788 | 0.6995 | 0.8122 | 3 |
| **Mean** | 0.7180 | 0.7764 | 0.6596 | 0.7800 | — |

indicates a greater probability of financial crises. This could be attributed to the increased interpretational burden imposed on information users by texts with lower readability, which obstructs a clear comprehension of the company's actual financial position and performance level [55]. Consequently, management tends to favor the disclosure of less readable text information to conceal negative news, such as poor performance.

**Analysis of empirical results of financial crisis early warning based on traditional financial indicators and MD&A text similarity indicator.** High MD&A text similarity indicates that listed companies engage in passive and perfunctory information disclosure, treating it as a mere formality, lacking objective analysis and timely updates on their actual operating situation during the reporting period, or showing limited progress compared to the past [59]. In the original data, ST-listed companies exhibit higher MD&A text similarity, with a mean value of 0.6377, while non-ST-listed companies have a mean value of 0.5666 for this indicator. To examine whether MD&A text similarity can enhance the effectiveness of traditional financial indicators in identifying companies' financial crisis, we utilize traditional financial indicators and MD&A text similarity indicator as input feature variables. The specific results are shown in in Table 7. By comparing the effectiveness of thirteen machine learning models based on traditional financial indicators, we can find that all models exhibit a significant improvement in accuracy, specificity, sensitivity, and AUC, with increases exceeding 0.04. Besides, the overall mean values all substantially increased. Specifically, the mean accuracy, mean specificity and mean AUC increase by more than 0.09, with values of 0.0924, 0.1189 and 0.0959 respectively. The mean value of sensitivity shows a relatively lower increase but remains at a high level of 0.0658.

The significant increase in numbers indicates that the MD&A text similarity indicator is effective for improving financial crisis early warning of listed companies. The results observed in the Chinese market align with the previous research findings of scholars. High similarity in information is often associated with inadequate corporate governance, unreliable accounting information quality, and poor operational performance.

**Analysis of empirical results of financial crisis early warning based on traditional financial indicators and the combination of MD&A text-linguistic feature indicators with enhanced warning effects.** The empirical results in the previous sections show that MD&A text readability and similarity indicators can significantly enhance the early warning effect of

**Table 7. Early warning effect based on traditional financial indicators and MD&A text similarity indicator.**

|  | Accuracy | Specificity | Sensitivity | AUC | Rank |
|---|---|---|---|---|---|
| Logistic regression | 0.6947 | 0.7067 | 0.6827 | 0.7359 | 10 |
| Ridge regression | 0.6827 | 0.6971 | 0.6683 | 0.7221 | 12 |
| Lasso regression | 0.6839 | 0.7019 | 0.6659 | 0.7274 | 11 |
| SVM | 0.6875 | 0.7188 | 0.6563 | 0.7218 | 13 |
| BPNN | 0.7115 | 0.7981 | 0.6250 | 0.7646 | 9 |
| DT | 0.7332 | 0.7957 | 0.6707 | 0.8077 | 7 |
| GBDT | 0.7524 | 0.8173 | 0.6875 | 0.8259 | 4 |
| CatBoost | 0.7548 | 0.8389 | 0.6707 | 0.8279 | 2 |
| XGBoost | 0.7464 | 0.7981 | 0.6947 | 0.8213 | 5 |
| LightGBM | 0.7404 | 0.8029 | 0.6779 | 0.8181 | 6 |
| AdaBoost | 0.7175 | 0.7404 | 0.6947 | 0.7854 | 8 |
| RF | 0.7596 | 0.8173 | 0.7019 | 0.8362 | 1 |
| Bagging | 0.7428 | 0.7692 | 0.7163 | 0.8272 | 3 |
| **Mean** | 0.7237 | 0.7694 | 0.6779 | 0.7863 | — |

**Table 8. Early warning effect based on traditional financial indicators and the combination of MD&A text readability and similarity indicators.**

|  | Accuracy | Specificity | Sensitivity | AUC | Rank |
|---|---|---|---|---|---|
| Logistic regression | 0.7163 | 0.7476 | 0.6851 | 0.7708 | 11 |
| Ridge regression | 0.7163 | 0.7572 | 0.6755 | 0.7624 | 13 |
| Lasso regression | 0.7151 | 0.7428 | 0.6875 | 0.7651 | 12 |
| SVM | 0.7356 | 0.7837 | 0.6875 | 0.7734 | 10 |
| BPNN | 0.8077 | 0.8870 | 0.7284 | 0.8350 | 9 |
| DT | 0.8149 | 0.8750 | 0.7548 | 0.8776 | 7 |
| GBDT | 0.8510 | 0.9159 | 0.7861 | 0.8921 | 3 |
| CatBoost | 0.8522 | 0.9255 | 0.7788 | 0.8974 | 1 |
| XGBoost | 0.8486 | 0.9014 | 0.7957 | 0.8919 | 4 |
| LightGBM | 0.8474 | 0.9014 | 0.7933 | 0.8913 | 5 |
| AdaBoost | 0.7740 | 0.8005 | 0.7476 | 0.8365 | 8 |
| RF | 0.8486 | 0.8990 | 0.7981 | 0.8964 | 2 |
| Bagging | 0.8149 | 0.8389 | 0.7909 | 0.8858 | 6 |
| **Mean** | 0.7956 | 0.8443 | 0.7469 | 0.8443 | — |

companies' financial crises. Therefore, in this section, we combine MD&A text readability and similarity indicators together with traditional financial indicators as input feature variables to examine the early warning effect of thirteen machine learning models. The objective is to assess the improvement of the combination of MD&A text-linguistic feature indicators with early warning effect enhancement and traditional financial indicators on the early warning effect. The specific results are shown in in Table 8.

Compared to the early warning effects demonstrated by the five empirical results shown previously, significant increases appear in the mean of accuracy, specificity, sensitivity, and AUC for the machine learning models of 0.1643, 0.1938, 0.1348, and 0.1539, respectively. In addition, all thirteen machine learning models exhibit significant increases in all four evaluation indexes. These findings indicate that the combination of MD&A text-linguistic feature indicators, namely readability and similarity, is much more effective in enhancing the financial crisis early warning of listed companies.

In addition, in the previous analysis, we failed to find significant effects of machine learning models incorporating MD&A text tone indicator for early warning. In this section, based on the analysis presented in Table 8, MD&A text tone indicator are added. In other words, MD&A text readability, similarity and text tone indicators are used as input feature variables along with traditional financial indicators. If the model incorporating the three text-linguistic feature indicators dose not demonstrate a significant improvement in accuracy, specificity, sensitivity, and AUC compared to the model with only text readability and similarity indicators, it would provide stronger evidence that the MD&A text tone indicator does not significantly enhance the financial crisis early warning of listed companies.

The specific results are shown in Table 9. We can see that the accuracy, specificity, sensitivity, and AUC of the thirteen machine learning models are generally consistent with the values reported in Table 8. Thus, based on the available evidence, it can be concluded that the MD&A text tone indicator does not have a prominent enhancement effect on the financial crisis early warning of listed companies.

**Comparative analysis of the performance of each machine learning model for sample recognition.** In this section, we calculate the average values of early warning effect evaluation indexes for thirteen machine learning models using seven different sets of input feature variable. Each effect evaluation index is ranked to analyze and compare the performance of each

**Table 9. Early warning effect based on traditional financial indicators and the combination of MD&A text tone, readability and similarity indicators.**

|  | Accuracy | Specificity | Sensitivity | AUC | Rank |
|---|---|---|---|---|---|
| Logistic regression | 0.7284 | 0.7644 | 0.6923 | 0.7812 | 11 |
| Ridge regression | 0.7175 | 0.7500 | 0.6851 | 0.7730 | 13 |
| Lasso regression | 0.7296 | 0.7596 | 0.6995 | 0.7763 | 12 |
| SVM | 0.7296 | 0.7572 | 0.7019 | 0.7839 | 10 |
| BPNN | 0.7704 | 0.8510 | 0.6899 | 0.8300 | 9 |
| DT | 0.8269 | 0.8846 | 0.7692 | 0.8873 | 6 |
| GBDT | 0.8413 | 0.9038 | 0.7788 | 0.8979 | 3 |
| CatBoost | 0.8438 | 0.9183 | 0.7692 | 0.9001 | 1 |
| XGBoost | 0.8401 | 0.8918 | 0.7885 | 0.8958 | 4 |
| LightGBM | 0.8413 | 0.8942 | 0.7885 | 0.8948 | 5 |
| AdaBoost | 0.7813 | 0.8053 | 0.7572 | 0.8467 | 8 |
| RF | 0.8486 | 0.9014 | 0.7957 | 0.9000 | 2 |
| Bagging | 0.8137 | 0.8269 | 0.8005 | 0.8839 | 7 |
| **Mean** | 0.7933 | 0.8391 | 0.7474 | 0.8501 | — |

machine learning model in sample recognition. The specific results are shown in Table 10, indicating that 6 models achieve accuracy scores above the mean value, 8 models exhibit specificity scores above the mean value, 7 models demonstrate sensitivity scores above the mean value, and 6 models attain AUC scores above the mean value.

There are six models, namely RF, Bagging, CatBoost, GBDT, XGBoost and LightGBM, that surpass the mean value for all four effectiveness evaluation indexes. These six models consistently rank among the top 6 in terms of AUC, sensitivity, and accuracy. Except for Bagging, the remaining five models also rank in the top 6 for specificity. Despite Bagging ranking 8th in specificity, it still maintains a prominent position. The empirical results from the previous section show that the top 5 models ranked in terms of AUC in each result are produced in these six machine learning models, so these six models have strong predictive performance on all seven sets of input features. In contrast, logistic regression, ridge regression, lasso regression and SVM models exhibit poor performance in sample identification. Additionally, while DT

**Table 10. Mean values and ranking of early warning evaluation indicators of 13 machine learning models based on seven different sets of input feature variables.**

|  | Accuracy | Rank | Specificity | Rank | Sensitivity | Rank | AUC | Rank |
|---|---|---|---|---|---|---|---|---|
| Logistic regression | 0.6834 | 10 | 0.7095 | 10 | 0.6573 | 8 | 0.7292 | 10 |
| Ridge regression | 0.6702 | 13 | 0.6975 | 13 | 0.6429 | 12 | 0.7173 | 13 |
| Lasso regression | 0.6760 | 11 | 0.6981 | 12 | 0.6538 | 9 | 0.7215 | 11 |
| SVM | 0.6743 | 12 | 0.6992 | 11 | 0.6494 | 10 | 0.7191 | 12 |
| BPNN | 0.6990 | 9 | 0.7534 | 7 | 0.6446 | 11 | 0.7462 | 9 |
| DT | 0.7007 | 7 | 0.7610 | 4 | 0.6405 | 13 | 0.7612 | 7 |
| GBDT | 0.7296 | 3 | 0.7668 | 3 | 0.6923 | 4 | 0.7893 | 4 |
| CatBoost | 0.7347 | 2 | 0.7888 | 1 | 0.6806 | 5 | 0.7964 | 2 |
| XGBoost | 0.7249 | 4 | 0.7545 | 6 | 0.6954 | 3 | 0.7879 | 5 |
| LightGBM | 0.7174 | 6 | 0.7552 | 5 | 0.6796 | 6 | 0.7836 | 6 |
| AdaBoost | 0.6997 | 8 | 0.7229 | 9 | 0.6765 | 7 | 0.7606 | 8 |
| RF | 0.7375 | 1 | 0.7716 | 2 | 0.7033 | 1 | 0.8010 | 1 |
| Bagging | 0.7213 | 5 | 0.7414 | 8 | 0.7012 | 2 | 0.7920 | 3 |
| **Mean** | 0.7053 | — | 0.7400 | — | 0.6706 | — | 0.7619 | — |

ranks 4th in specificity, it ranks lowest in sensitivity. Given that our primary purpose is to identify ST-listed companies, this model is not the focus of our attention.

## Further discussion

The empirical analysis results show that the inclusion of MD&A text readability and similarity indicators in models enhances the effectiveness of traditional financial indicators for identifying financial crises in listed companies, offering incremental information. Nevertheless, we fail to find a similar enhancement effect of models incorporating MD&A text tone and forward-looking indicators in identifying financial crises.

Firstly, these results suggest that information on text readability and similarity contributes incremental information value, aligning with existing research linking text readability and similarity to company performance [15, 36]. Underperforming companies may intentionally obscure negative operational information and conceal unfavorable news by reducing readability [13, 14]. Moreover, such companies usually exhibit negative disclosure attitudes and limited operational progress, resulting in higher MD&A text similarity [15, 59]. In our study, we explore the early warning effect of these two indicators on financial crises using MD&A texts from Chinese listed companies. Furthermore, we find that text readability and similarity indicators can serve as early warning signals for financial crises, where lower readability and higher similarity indicate a higher likelihood of financial crises.

In addition, the empirical results in this study show that the text tone and forward-looking indicators of ST and non-ST-listed companies are indistinguishable from each other. Both are redundant rather than incremental information. This study confirms that management could manipulate the tone and forward-looking aspects of textual information. The presence of "language inflation" within strategic statements undermines their incremental information value [9, 10, 13].

Regarding the early warning effect of thirteen machine learning models, it is evident that six models, namely RF, Bagging, CatBoost, GBDT, XGBoost and LightGBM, exhibit strong predictive performance across seven sets of input feature variables. These findings provide an empirical foundation for further research and practical application.

## Conclusions

This study focus on Chinese A-share listed companies during the period of 2011–2020. The MD&A section of their annual reports is collected using web crawler technology. Subsequently, natural language processing technology is used to extract text tone, forward-looking, readability and similarity indicators from the MD&A texts. These text-linguistic feature indicators are then combined with traditional financial indicators to examine the incremental information value of MD&A text-linguistic features in predicting the financial crisis of listed companies. The conclusions are as follows: (1) MD&A text tone and forward-looking indicators do not have a significant effect on the enhancement of listed companies' financial crisis warning effect, even the latter brings some noise; (2) MD&A text readability and similarity indicators both provide incremental information for listed companies' financial crisis early warning and improve the accuracy of that early warning; (3) the combination of MD&A text readability and similarity indicators proves more effective in enhancing the early warning of listed companies' financial crises; (4) RF, Bagging, CatBoost, GBDT, XGBoost and LightGBM maintain stable and accurate sample recognition under seven different sets of input feature variables. These six models can be considered relatively relatively optimal classifiers for early warning of financial crisis in listed companies.

According to the above research results, the following inspiration could be drawn: (1) MD&A text tone and forward-looking indicators do not significantly improve the financial crisis warning effect of listed companies. This can be attributed to the challenge of accurately detecting financial crisis signals amid the strategic statements that may camouflage the actual situation. Therefore, the utilization of text tone and forward-looking indicators should be approached with caution in financial crisis early warning models; (2) MD&A text readability and similarity indicators can improve the early warning effect of models. This finding serves as a valuable supplement to relying solely on traditional financial indicators for crisis early warning, thereby expanding the range of financial crisis early warning indicators for listed companies and enabling more accurate predictions of their financial crises; (3) the combination of MD&A text readability and similarity indicators is more effective in improving the financial crisis early warning effect of listed companies. This means that when constructing a financial crisis early warning model, it is crucial to comprehensively consider these two MD&A text-linguistic feature indicators in order to achieve more accurate early warning results; (4) the accuracy of financial crisis early warning greatly depends on selecting an appropriate classifier. In practical applications, the most suitable model can be chosen for prediction based on the dataset characteristics and prediction requirements. Viable options include RF, Bagging, CatBoost, GBDT, XGBoost and LightGBM. To sum up, the key to early warning of financial crisis of listed companies lies in selecting an appropriate classifier and comprehensively considering multiple input feature variables. These inspirations can assist regulators and stakeholders in effectively predicting financial crises and providing robust decision support.

However, there are limitations in this study, which will guide our further research direction. These limitations include the following: (1) the sample dataset is not divided based on the industry of the listed companies. Constructing a financial crisis early warning model for listed companies using an industry-specific sample dataset could better align with industry characteristics; (2) this study is confined to listed companies, and further research could explore the applicability of the early warning model to non-listed companies; (3) there is a lack of research on other texts, such as audit reports of listed companies and investors' comments on social media platforms and so on. Exploring these texts could provide valuable insights for financial crisis early warning.

## Supporting information

**S1 Appendix.**
(DOCX)

## Author Contributions

**Conceptualization:** Huayong Niu.

**Methodology:** Zhishuo Zhang.

**Writing – original draft:** Zhishuo Zhang, Xinran Liu.

**Writing – review & editing:** Zhishuo Zhang, Xinran Liu, Huayong Niu.

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
