## [Decision Letter · Decision Letter 0]

11 Jul 2023

PONE-D-23-13667Financial Crisis Early Warning of Chinese Listed Companies Based on MD&A Text-linguistic Feature IndicatorsPLOS ONE

Dear Dr. Niu,

Thank you for submitting your manuscript to PLOS ONE. After careful consideration, we feel that it has merit but does not fully meet PLOS ONE’s publication criteria as it currently stands. Therefore, we invite you to submit a revised version of the manuscript that addresses the points raised during the review process.

We look forward to receiving your revised manuscript.

Kind regards,

Normaizatul Akma Saidi, Ph.D

Academic Editor

PLOS ONE

“Funder: Zhishuo Zhang: Methodology, Writing—original draft preparation, Writing—review and editing. This work was supported by Beijing Foreign Studies University Double First Class Major Landmark Project [grant number 2022SYLZD001]; the Fundamental Research Funds for the Central Universities [grant number 2023JX095]; and Beijing Foreign Studies University G20 Research Center Project [grant number G20ZX20223003].”

“This work was supported by Beijing Foreign Studies University Double First Class Major Landmark Project [grant number 2022SYLZD001]; the Fundamental Research Funds for the Central Universities [grant number 2023JX095]; and Beijing Foreign Studies University G20 Research Center Project [grant number G20ZX20223003].”

“Funder: Zhishuo Zhang: Methodology, Writing—original draft preparation, Writing—review and editing. This work was supported by Beijing Foreign Studies University Double First Class Major Landmark Project [grant number 2022SYLZD001]; the Fundamental Research Funds for the Central Universities [grant number 2023JX095]; and Beijing Foreign Studies University G20 Research Center Project [grant number G20ZX20223003].”

6. Please amend either the abstract on the online submission form (via Edit Submission) or the abstract in the manuscript so that they are identical.

Reviewers' comments:

Reviewer's Responses to Questions

**Comments to the Author**

1. Is the manuscript technically sound, and do the data support the conclusions?

Reviewer #1: Partly

Reviewer #2: Yes

2. Has the statistical analysis been performed appropriately and rigorously? 

Reviewer #1: No

Reviewer #2: Yes

3. Have the authors made all data underlying the findings in their manuscript fully available?

Reviewer #1: No

Reviewer #2: Yes

4. Is the manuscript presented in an intelligible fashion and written in standard English?

Reviewer #1: Yes

Reviewer #2: Yes

5. Review Comments to the Author

Reviewer #1: 1，The authors normalized the data and adopted a 10-fold cross-validation method to enhance the robustness of the study. The results also support the study's conclusions.

2，The authors did not perform a descriptive analysis of traditional financial indicators. A descriptive analysis of the indicators should be added in subsequent revisions.

3，I do not see the data submitted by the authors.

Reviewer #2: In this manuscript, the authors study the financial crisis early warning of Chinese listed companies using many kinds of technologies. This manuscript is rich in content and the authors have done a lot of work. However, I have some concerns about the work that the authors should clarify:

(1) Is this manuscript a review paper? There are no Innovation in methods.

(2) Although the manuscript is lengthy, there is relatively little original content created by the authors.

(3) There is less introduction to web crawler technology in the manuscript.

(4) The annotation of the experimental results in the text is not clear, and it cannot be concluded which method is more superior. The conclusions obtained through experiments are limited in content and do not show particularly valuable conclusions.

(5) The superiority of machine learning methods cannot be highlighted without comparison with other types of methods.

(6) Some variables in the text are formatted in italics, while others are formatted in normal font. It should be unified. Please further standardize the format of the manuscript.

6. PLOS authors have the option to publish the peer review history of their article (what does this mean?). If published, this will include your full peer review and any attached files.

Reviewer #1: **Yes: **Zitian Fu

Reviewer #2: No

---

## [Author Response · Author response to Decision Letter 0]

23 Jul 2023

Response to Reviewer 1

Point 1: Abstract：

(1)In the first sentence of the abstract, the author should provide a general overview of the background and importance of the study.

(2) The part of the abstract that introduces the content of the article lacks logic. The author can use such logical words: "first", "on this basis", etc., to make the logic of the front and back more closely, and to help clearly express the point of view that the article wants to convey.

Response 1: Thank you for your valuable suggestion. 

(1)Based on your suggestions, we have added the following statement in the manuscript: Nowadays, the international situation is severe and complex, and the structural problems of the Chinese economy are prominent, which makes the financial risks of Chinese listed companies rise continuously. Thus it is of great practical significance to provide effective financial crisis early warnings for listed companies.

(2)We further added logical words such as "first, and then, through the above research, in general" to enhance the logic of the content, and the revised content is as follows:

In this paper, we first use web crawler technology and natural language processing technology to measure four text-linguistic features in the Management Discussion and Analysis (MD&A) section of the annual financial reports of listed companies in China from 2011 to 2020. Those text-linguistic features are text tone, forward-looking, readability and similarity. And then we combine them with traditional financial indicators respectively, use thirteen mainstream machine learning models to comparatively study different combinations of those indicators and machine learning models for the effectiveness of the early warning on listed companies’ financial crises. Through the above research, the empirical results show that MD&A text readability and similarity indicators can provide more incremental information for prediction models, while text tone and forward-looking indicators are converses. The latter two are easily modified by management, so bring more noise to the prediction. Finally, by comparing the early warning effects of thirteen machine learning models, it is found that RF, Bagging, CatBoost, GBDT, XGBoost and LightGBM models maintain stable and accurate sample recognition ability. In general, this paper constructs a more effective financial crisis early warning model by exploring the MD&A text-linguistic feature indicators, which provides a new research perspective for further research in this field.

Point 2: Introduction

(1)It is necessary to cite literature to enhance persuasiveness. For example, which literature is referred to in this sentence: "Management's Discussion and Analysis (MD&A) contains further interpretation and analysis of the listed company's annual financial statements by the management"

(2)In the part of introducing the background of the article, there is a lack of logic between sentences, and more consideration needs to be given to how to connect the sentences well.

(3)Please describe the deficiencies of existing research in detail, and add corresponding references.

(4)The contribution part of the article should be compared with previous research to enhance persuasiveness.

Response 2: (1) According to your suggestion, we enhance persuasiveness by citing some literature, which is cited below:

Epstein, Barry J., and Eva K. Jermakowicz. WILEY Interpretation and Application of International Financial Reporting Standards 2010. John Wiley & Sons, 2010.

Zhao Q, Xu W, Ji Y. Predicting financial distress of Chinese listed companies using machine learning: To what extent does textual disclosure matter? International Review of Financial Analysis, 2023: 102770.

Ren C, Lu X. A multi-dimensional analysis of the management's discussion and analysis narratives in Chinese and American corporate annual reports. English for Specific Purposes, 2021, 62: 84-99.

(2) We have added some logical words and polished the corresponding language to improve the logic of the "introduction", and the modified parts used tracked changes in the manuscript.

(3) The main shortcomings of the existing studies are: (1) MD&A text tone and forward-looking information are susceptible to whitewashing, and there is controversy over whether text tone and forward-looking information have incremental information value [9-14]; (2) at present, there is no study that uses the readability of MD&A texts for early warning of company financial crises; (3) there is no relevant research on the Chinese market using the MD&A text similarity indicator[15]; (4) the financial crisis early warning methods for listed companies used in existing studies are mostly comparative studies using a single model or a few models, and there are relatively few comparative studies using multiple machine learning models [16-19].

(4) Based on your suggestions, the original contribution section of this paper has been further modified, and the contents of the modified contribution section are as follows:

The main contributions of this paper are: (1) Based on the same sample dataset, this paper measures the enhancement effect of each text-linguistic feature indicator in MD&A text on the financial crisis early warning of listed companies, which can provide a clearer picture of the incremental information contained in different text-linguistic features, and make up for the shortcomings of the existing research; (2) finding that the text readability and similarity indicators in MD&A text-linguistic features have incremental information, which is conducive to enhancing the accuracy of early warning models for listed companies' financial crises; (3) exploring the financial crisis early warning effectiveness of listed companies with the combination of MD&A text-linguistic feature indicators with enhanced early warning effect, which enriches the relevant research perspectives; (4) comparing thirteen mainstream machine learning models, and finding six of them with better and stable early warning effects, which provides a reference for subsequent model users.

Point 3: Empirical Analysis

The authors did not perform a descriptive analysis of traditional financial indicators. A descriptive analysis of the indicators should be added in subsequent revisions

Response 3: We added descriptive statistics in the chapter "Data Processing and Descriptive Statistics", and the descriptive statistics of traditional financial indicators and text-linguistic feature indicators are shown in Table 2. We uploaded the raw data to the following public repository: https://github.com/zzs0216/Financial-Crisis-Early-Warning-of-Chinese-Listed-Companies/tree/main.

Table 2. Descriptive statistics of traditional financial indicators and text-linguistic feature indicators

Variable Obs Mean Std. Dev. Min Max

Earnings per share 832 0.14 0.57 -5.88 4.49 

Operating profit ratio 832 -0.21 4.18 -116.76 13.97 

Debt assets ratio 832 0.56 0.43 0.02 8.00 

Current ratio 832 1.45 1.57 0.00 21.12 

Cash coverage ratio 832 0.09 0.52 -9.22 4.31 

Interest cover 832 48.67 1,244.86 -1,471.08 35,725.11 

Accounts receivable turnover ratio 832 329.61 4,671.90 0.27 124,214.10 

Sales-to-cash ratio 832 1.02 0.90 0.12 25.15 

Operating cash per share 832 0.22 0.85 -6.65 9.42 

Net profit growth rate 832 -131.71 1,301.70 -23,572.69 13,737.70 

Tone Indicator 832 0.40 0.17 -0.22 1.00 

Forward-Looking Indicator 832 0.01 0.00 0.00 0.02 

Readability Indicator 832 -21.88 4.26 -42.82 -11.20 

Similarity Indicator 832 0.60 0.16 0.03 0.97 

ST (Not St is 1, St is 2) 832 1.50 0.50 1 2

Point 4: Related Studies:

The literature review part is to summarize previous research. The literature review section does not need to elaborate on the research content of this paper again.

It is suggested that the author remove what was studied in this article.

Response 4: Thank you for your valuable suggestion. We have removed what was studied in this article in "Related Studies".

Point 5: Conclusions:

Please describe in detail what the inspiration of this article is.

Response 5: Based on your suggestions, we have added the description in detail what the inspirations of this paper is, as followed:

According to the above research results, the following inspiration could be drawn: (1) the addition of MD&A text tone and forward-looking indicators does not significantly improve the financial crisis warning effect of listed companies, because it is difficult to accurately capture the signal of financial crisis under the whitewash of strategic statements. Therefore, in the financial crisis early warning model, the use of text tone and forward-looking indicators needs to be carefully considered; (2) MD&A text readability and similarity indicators can improve the early warning effect of models based on traditional financial indicators. This finding is a useful supplement to relying solely on traditional financial indicators for crisis early warning, further expanding the financial crisis early warning indicators of listed companies, which can more accurately predict the financial crisis of listed companies; (3) the combination of MD&A text readability and similarity indicators is more effective in improving the financial crisis early warning effect of listed companies. This means that when building a financial crisis early warning model, these two MD&A text-linguistic feature indicators should be considered comprehensively to obtain more accurate early warning results; (4) selecting an appropriate classifier is crucial to the accuracy of financial crisis early warning. In practical applications, the most suitable model can be selected for prediction according to the characteristics of the data set and the prediction requirements. RF, Bagging, CatBoost, GBDT, XGBoost and LightGBM are all viable options. To sum up, choosing an appropriate classifier and comprehensively considering multiple input feature variables are the key to early warning of financial crisis of listed companies. These inspirations can help regulators and stakeholders to better warn financial crises in practice and can provide strong decision support.

Response to Reviewer 2

Point 1: Is this manuscript a review paper? There are no Innovation in methods.

Response 1: First, this manuscript is a research article, not a review paper. And then, Please allow me to make the following explanation:

Financial crisis early warning of listed companies usually starts with selecting some predictive indicators that are indicative of whether a financial crisis will occur in the future, and then based on these predictive indicators, applying statistical analysis or machine learning methods to predict the financial crisis. According to this process, the crisis early warning effect depends on two aspects: on the one hand, whether the predictive indicators can scientifically and reasonably provide a comprehensive evaluation of the overall situation of listed companies, and on the other hand, on the selection of forecasting method technology. In other words, the predictive indicators determine the upper limit of prediction accuracy, and the prediction method is to keep approaching this upper limit. Therefore, in terms of predictive indicators, the focus of this paper is to explore the incremental information of MD&A text-linguistic feature indicators for the financial crisis early warning of listed companies in order to further improve the upper limit of prediction accuracy; while in terms of methodology, although we did not innovate existing modeling methods, we explored the warning effects of 13 mainstream machine learning models, bridging the gap of multi-model comparative studies in this area.

Point 2: Although the manuscript is lengthy, there is relatively little original content created by the authors.

Response 2: 

It is possible that our presentation in the manuscript was not clear enough to highlight our innovations. Therefore, we have added the shortcomings of existing studies and refined the contributions of this research in section "Introduction" as follows:

The main shortcomings of the existing studies are: (1) MD&A text tone and forward-looking information are susceptible to whitewashing, and there is controversy over whether text tone and forward-looking information have incremental information value [9-14]; (2) at present, there is no study that uses the readability of MD&A texts for early warning of company financial crises; (3) there is no relevant research on the Chinese market using the MD&A text similarity indicator[15]; (4) the financial crisis early warning methods for listed companies used in existing studies are mostly comparative studies using a single model or a few models, and there are relatively few comparative studies using multiple machine learning models [16-19].

The main contributions of this paper are: (1) Based on the same sample dataset, this paper measures the enhancement effect of each text-linguistic feature indicator in MD&A text on the financial crisis early warning of listed companies, which can provide a clearer picture of the incremental information contained in different text-linguistic features, and make up for the shortcomings of the existing research; (2) finding that the text readability and similarity indicators in MD&A text-linguistic features have incremental information, which is conducive to enhancing the accuracy of early warning models for listed companies' financial crises; (3) exploring the financial crisis early warning effectiveness of listed companies with the combination of MD&A text-linguistic feature indicators with enhanced early warning effect, which enriches the relevant research perspectives; (4) comparing thirteen mainstream machine learning models, and finding six of them with better and stable early warning effects, which provides a reference for subsequent model users.

Point 3: There is less introduction to web crawler technology in the manuscript.

Response 3: Thank you for your valuable suggestion. Considering that this article is not a technical article, the unoriginal web crawler technology is only a way to get the text, so it is not expanded to explain. In addition, considering that the existing manuscript already has a large amount of content, adding a detailed introduction to web crawler technology may lead to problems such as fragmented content and lack of focus, so we hope to have your approval and understanding.

Point 4: The annotation of the experimental results in the text is not clear, and it cannot be concluded which method is more superior. The conclusions obtained through experiments are limited in content and do not show particularly valuable conclusions.

Response 4: Allow us to explain further on this issue. Firstly, we introduce the evaluation indicators of the model early warning effect in the section "Machine Learning Model Early Warning Effect Evaluation Indicator", which are Accuracy, Sensitivity, Specificity and AUC value, and further introduce the meaning of the indicators and the specific measurement formula. In the subsequent empirical studies, the results of each sub-empirical study demonstrated the above four indicators, which are also specified in the empirical analysis. Specifically, when evaluating the financial crisis early warning effect of listed companies incorporating MD&A text-linguistic feature indicators, we use the average scores of the above four evaluation indicators of the 13 machine learning models to indicate the financial crisis early warning effect, and compare them with the effect of only the traditional financial indicators, in order to derive the value of the incremental information of the MD&A textual-linguistic feature indicators. In addition, we also compare the early warning effect of 13 models on different indicators of the same sample dataset to derive the recognition performance of 13 machine learning models for the samples. And the scores and rankings of the four evaluation indicators are shown in "Table A8. Mean values and ranking of early warning effect evaluation indicators of 13 machine learning models based on seven different sets of input feature variables", while are also discussed in section "Comparative Analysis of the Performance of Each Machine Learning Model for Sample Recognition".

Based on the above research, the paper draws the following conclusions:

(1) the addition of MD&A text tone and forward-looking indicators brings some noise and does not have a significant effect on the enhancement of listed companies' financial crisis warning effect; (2) MD&A text readability and similarity indicators both provide incremental information for listed companies' financial crisis early warning and improve the accuracy of that early warning; (3) the combination of text-linguistic feature indicators composed of MD&A text readability and similarity indicators is more effective in enhancing the early warning of listed companies' financial crises; (4) under seven different sets of input feature variables, RF, Bagging, CatBoost, GBDT, XGBoost and LightGBM maintain stable and accurate sample recognition, and these six models can be used as relatively optimal classifiers for listed companies' financial crisis early warning.

Point 5: The superiority of machine learning methods cannot be highlighted without comparison with other types of methods.

Response 5: 

In this paper, based on the existing literature, we know that machine learning is a mainstream approach in the field of prediction with good predictive results.

Besides, this paper is not to emphasize that machine learning methods are superior to everything. After all, any method has its limitations. The focus of this paper is on the issue of the incremental value of information of MD&A text-linguistic feature indicators. Based on that, the issue of the superiority of 13 machine learning methods is compared.

Point 6: Some variables in the text are formatted in italics, while others are formatted in normal font. It should be unified. Please further standardize the format of the manuscript.

Response 6: Thank you for your valuable suggestion. The manuscript has been further adapted to the issues you have raised, as well as other formatting issues, in accordance with the style requirements of the journal.

---

## [Decision Letter · Decision Letter 1]

15 Aug 2023

PONE-D-23-13667R1Financial Crisis Early Warning of Chinese Listed Companies Based on MD&A Text-linguistic Feature IndicatorsPLOS ONE

Dear Dr. Huayong Niu,

Thank you for submitting your manuscript to PLOS ONE. After careful consideration, we feel that it has merit but does not fully meet PLOS ONE’s publication criteria as it currently stands. Therefore, we invite you to submit a revised version of the manuscript that addresses the points raised during the review process.

ACADEMIC EDITOR: Thank you for revising your manuscript. There is one minor last revision needed before moving forward. Looking forward to receiving your revised submission.

We look forward to receiving your revised manuscript.

Kind regards,

Normaizatul Akma Saidi, Ph.D

Academic Editor

PLOS ONE

Journal Requirements:

Reviewers' comments:

Reviewer's Responses to Questions

**Comments to the Author**

1. If the authors have adequately addressed your comments raised in a previous round of review and you feel that this manuscript is now acceptable for publication, you may indicate that here to bypass the “Comments to the Author” section, enter your conflict of interest statement in the “Confidential to Editor” section, and submit your "Accept" recommendation.

Reviewer #1: (No Response)

Reviewer #2: All comments have been addressed

Reviewer #3: (No Response)

Reviewer #4: (No Response)

2. Is the manuscript technically sound, and do the data support the conclusions?

Reviewer #1: Yes

Reviewer #2: Yes

Reviewer #3: Partly

Reviewer #4: No

3. Has the statistical analysis been performed appropriately and rigorously? 

Reviewer #1: Yes

Reviewer #2: Yes

Reviewer #3: N/A

Reviewer #4: No

4. Have the authors made all data underlying the findings in their manuscript fully available?

Reviewer #1: Yes

Reviewer #2: Yes

Reviewer #3: Yes

Reviewer #4: Yes

5. Is the manuscript presented in an intelligible fashion and written in standard English?

Reviewer #1: Yes

Reviewer #2: Yes

Reviewer #3: Yes

Reviewer #4: No

6. Review Comments to the Author

Reviewer #1: The data for the paper has been submitted. The model process of the paper is introduced in more detail. Further improvement is needed in English expression.

Reviewer #2: The authors have adequately addressed my comments raised in a previous round of review and I feel that this manuscript is now acceptable for publication.

Reviewer #3: Point 1:

This study has relatively important practical value. It is recommended that the authors double-check the grammar and spelling throughout this manuscript to ensure that the quality of this manuscript is not affected by those minor issues.

For example, Introduction, paragraph 4, “select 832 listed companies of Chinese RMB common stocks (A shares) during 2012-2020 as the sample”. But in fact, the sample period used in this manuscript is 2011-2020.

Related Studies, paragraph 7, “so cannot be be considered as having incremental information value”.

Point 2: Structure of this manuscript.

It is suggested that the Machine Learning Models section be placed in the appendix, while the tables of empirical results be placed in the main text. On the one hand, these two sections are nearly the same length in terms of pages. On the other hand, the result section is the focus of readers’ attention and the authors’ main contribution. However, Machine Learning Models is not the contribution or innovation of this study; rather, it is an introduction of supplementary knowledge on the methodology.

Point 3: Empirical Analysis section.

It is recommended that the authors explicitly indicate the software or tools used in this manuscript’s empirical analysis, including the NLP and ML.

Point 4: Improvement of the tables in the results section.

Comparing Tables A1 and A2, it is found that the ranking (in the Rank column) is not based on the average or total value of the four evaluation indicators (Accuracy, Specificity, Sensitivity, and AUC). It is suggested that the authors clearly state the ranking criteria for the models in this manuscript. In addition, the authors should add a column of data for the specific ranking indicators to the current tables, instead of simply listing the ranking results. This will help readers understand more intuitively the specific differences between each model.

For example, in Table A1, the average of the four evaluation indicators of the Logistic Regression model (0.6586) is higher than that of the Bagging model (0.6572), but the Rank of the Logistic Regression model is 6, whereas the Rank of the Bagging model is 2. It makes readers confused.

Point 5:

In the section of Analysis of Empirical Results of Financial Crisis Early Warning Based on Traditional Financial Indicators and MD&A Text Tone Indicator, there is a statement, “Although there is a slight increase in the mean value, there is still a decrease in some of the evaluation indicators measured by some models”. However, after comparing Tables A1 and A2, it is found that only two models (SVM and AdaBoost) have a decrease in evaluation indicators of specificity and AUC, while the rest have an increase. This is not a sufficient reason for the authors’ conclusion that the MD&A Text Tone Indicator is noisy. It is recommended that the authors strengthen their argument in this paragraph.

Reviewer #4: 2. Is the manuscript technically sound, and do the data support the conclusions?

The manuscript presents an in-depth investigation into predicting financial crises in Chinese A-share listed companies, employing MD&A text readability, similarity, tone, and forward-looking indicators from 2011-2020. The authors claim that text readability and similarity are valuable for identifying financial crises, supporting existing research connecting these indicators to firm performance. However, they found that tone and forward-looking indicators did not enhance the identification of financial crises, as they can be manipulated by management. Utilizing machine learning models such as RF, Bagging, CatBoost, GBDT, XGBoost, and LightGBM, they provided a nuanced understanding of how combining text-linguistic features with traditional financial indicators can enhance early warning systems for financial crises. There are certain areas where the technical soundness could be improved, and where the data may not fully support the conclusions:

• Methodology Clarity: The methodology section could provide more detailed information about the data preprocessing, feature extraction, and the selection process for the machine learning models. Examples or specific rationale for the chosen methods would enhance the clarity.

• Control and Replication: The manuscript lacks adequate information about controls, replication strategies, and potential confounding factors, such as industry-specific patterns, economic conditions, etc.

• Sample Size Justification: A clear justification of the sample size would add to the rigor of the study and ensure that the findings are robust and generalizable.

3. Has the statistical analysis been performed appropriately and rigorously?

The manuscript employs various machine learning models, but the statistical analysis needs further clarification:

• Model Validation: More details on cross-validation, hyperparameter tuning, and model evaluation metrics are needed to ensure the validity of the results.

• Comparison of Models: A thorough statistical comparison of the selected models against others would clarify why these specific models were chosen and how they outperform others.

4. Have the authors made all data underlying the findings in their manuscript fully available?

The manuscript does not provide complete information regarding the availability of the underlying data:

• Data Availability Statement: The Data Availability Statement should explicitly mention where the data can be accessed and whether there are any restrictions on data sharing.

• Data Description: A more thorough description of the dataset, including the web crawler technology used, and specific details about the extracted features, would enhance the reproducibility of the study.

5. Is the manuscript presented in an intelligible fashion and written in standard English?

The manuscript is mostly well-written but suffers from some grammatical errors and unclear expressions:

• Language and Grammar: Some sentences are complex and could be broken down into simpler structures to improve readability. A thorough proofreading is recommended.

• Figure and Table Descriptions: Additional explanations for figures and tables, along with concise captions, would enhance understanding.

6. Review Comments to the Author

This study presents a novel approach to early warning of financial crises in listed companies, leveraging text analysis and machine learning. While the concept is promising, there are several areas where the manuscript could be strengthened:

• Enhance the Technical Details: More information on methodology, statistical analysis, and data availability is essential to increase the rigor and reproducibility of the study.

• Consider Industry-Specific Analysis: Incorporating industry-specific characteristics might provide a more nuanced understanding and enhance the model's predictive power.

• Improve Language and Presentation: Proofreading for grammatical errors and enhancing figure and table descriptions will improve the manuscript's readability.

The addressed areas of improvement will not only enhance the technical soundness of the paper but also contribute to a broader understanding and applicability of the proposed methods. This research has the potential to make an impact, and I wish you all the best with its continued development.

7. PLOS authors have the option to publish the peer review history of their article (what does this mean?). If published, this will include your full peer review and any attached files.

Reviewer #1: No

Reviewer #2: No

Reviewer #3: No

Reviewer #4: No

---

## [Author Response · Author response to Decision Letter 1]

21 Aug 2023

Response to Reviewer 1 Comments

Point 1: The author's manuscript still lacks literature support. Some of the views summarized by the predecessors need literature support. For example“Financial crisis early warning generally starts with selecting predictors that are indicative of whether a crisis will occur, and then applying statistical analysis methods or machine learning methods to identify the financial crisis.”Including but not limited to this sentence requires literature support.

Response 1: Thank you for your valuable suggestion. We have provided relevant literature support for some of the views summarized by predecessors.

Point 2: “In terms of traditional financial indicators, this paper selects 10 specindicator descriptionific financial indicators as benchmarks from five aspects metioned above to measure the performance of listed companies. In terms of text-linguistic feature indicators, this paper forms MD&A text tone, forward-looking, readability and similarity indicators for financial crisis early warning.” This paragraph should be placed in the indicator description and construction section, not in the literature review. I have mentioned this suggestion before, please revise it carefully.

Response 2: We apologize for our oversight. We have removed this statement from the literature review section and included it in the indicator description section.

Point 3: In the "Traditional Financial Indicators" section, please explain the reasons for selecting the indicators, rather than just explaining which literatures you refer to.

Response 3: We appreciate your suggestion and have added the reasons for selecting the 10 traditional financial indicators in our manuscript. The specific additions are as follows:

The selection of these ten indicators as financial indicators for assessing the financial crisis early warning of listed companies is based on the following considerations: (1) Earnings per share: earnings per share refers to the indicator obtained by dividing a company's net profit by the weighted average number of shares issued. It is an important indicator for measuring the profitability associated with each share of a company's stock. A higher Earnings per share indicates a strong profitability of the company, while a lower Earnings per share may suggest a decline in profitability and the presence of financial risks. (2) Operating profit ratio: operating profit ratio refers to the ratio of operating profit to operating revenue, reflecting the company's ability to generate profits in its operations. A high operating profit ratio indicates strong profitability, while a low operating profit margin may suggest a decline in profitability and potential financial crisis. (3) Debt assets ratio: debt assets ratio represents the proportion of a company's assets that are financed through debt. It can reflect the company's debt risk and financial stability. A higher debt assets ratio indicates a higher level of debt risk for the company, while a lower debt assets ratio generally indicates higher financial stability for the company. (4) Current ratio: current ratio is a measure of a company's short-term debt-paying ability and liquidity risk. It assesses the company's ability to meet short-term liabilities by comparing its current assets to its current liabilities. A higher current ratio indicates that the company has sufficient current assets to cover short-term debts, while a lower current ratio may suggest insufficient liquidity and potential difficulties in meeting debt obligations, which can lead to a financial crisis. (5) Cash coverage ratio: cash coverage ratio measures the relationship between the net cash flow generated from operating activities and the ability to repay current liabilities in a listed company. It reflects the extent to which the company's cash flow covers short-term debt obligations. A higher cash coverage ratio indicates a stronger ability to repay debts using cash flow, while a lower ratio may suggest insufficient cash flow, making it difficult for the company to repay debts on time, potentially leading to a financial crisis. (6) Interest cover: interest cover is a measure of a company's ability to pay interest expenses. It represents the profitability that a company can pay interest expense before interest is paid. A higher interest cover indicates a stronger ability to cover interest payments, while a lower interest cover may suggest difficulties in paying interest, indicating that shareholder equity could be affected and indicating the possibility of a financial crisis. (7) Accounts receivable turnover ratio: accounts receivable turnover ratio reflects how quickly a company collects its accounts receivable and relates to its cash flow recovery capability as well as the risk associated with accounts receivable. A higher ratio indicates that the company is able to collect accounts receivable quickly and utilize funds efficiently. On the other hand, a lower ratio may suggest poor efficiency in collecting accounts receivable, indicating potential financial strain and bad debt risk. (8) Sales-to-cash ratio: sales-to-cash ratio reflects the proportion of cash received from the sale of goods or services to the total operating revenue of a company. A higher ratio indicates that the company can quickly convert sales revenue into cash income, while a lower ratio may suggest the company faces risks of delayed payments or insufficient liquidity. (9) Operating cash per share: operating cash per share represents the relationship between the cash flow generated from a company's operating activities and the total common shares outstanding at the end of the year. It measures the stability and sustainability of a company's operating cash flow and provides investors with more insightful financial information. (10) Net profit growth rate: net profit growth rate reflects changes in a company's profitability and reveals the trend of its profit-generating ability. A higher net profit growth rate has a positive impact on the financial stability and sustainability of a company, while a lower rate may indicate a decline in profitability and potential financial challenges.

In summary, these ten traditional financial indicators comprehensively consider profitability, solvency, asset operating efficiency, cash flow quality, and development quality of listed companies. They have a high level of reliability and significance in predicting financial crises, aiding in the assessment a company's financial condition and early warning of potential risks.

Point 4: The Results of Empirical Analysis section only illustrates the model results. Authors are invited to moderately analyze the results.

Response 4: According to your suggestion, we have enhanced the analysis in the Results of Empirical Analysis section.

Point 5: Please polish your English expression.

Response 5: Thank you for your valuable suggestion. We have polished the English expression. The modifications have been clearly marked in the article using tracked changes.

Response to Reviewer 2 Comments

Point 1: The authors have adequately addressed my comments raised in a previous round of review and I feel that this manuscript is now acceptable for publication.

Response 1: Thank you for your recognition. Your valuable suggestions during the article revision process have been highly beneficial to us. We sincerely appreciate your diligent peer review once again.

Response to Reviewer 3 Comments

Point 1: This study has relatively important practical value. It is recommended that the authors double-check the grammar and spelling throughout this manuscript to ensure that the quality of this manuscript is not affected by those minor issues.

For example, Introduction, paragraph 4, “select 832 listed companies of Chinese RMB common stocks (A shares) during 2012-2020 as the sample”. But in fact, the sample period used in this manuscript is 2011-2020.

Related Studies, paragraph 7, “so cannot be be considered as having incremental information value”.

Response 1: Thank you for your valuable suggestion. We have revised the questions you raised and conducted a thorough grammar and spelling check of the entire manuscript to ensure the quality of the manuscript is not affected by these issues.

Point 2: Structure of this manuscript.

It is suggested that the Machine Learning Models section be placed in the appendix, while the tables of empirical results be placed in the main text. On the one hand, these two sections are nearly the same length in terms of pages. On the other hand, the result section is the focus of readers’ attention and the authors’ main contribution. However, Machine Learning Models is not the contribution or innovation of this study; rather, it is an introduction of supplementary knowledge on the methodology.

Response 2: Based on your suggestion, we have relocated the Machine Learning Models section to the appendix, and moved the tables of empirical results to the main text.

Point 3: Empirical Analysis section.

It is recommended that the authors explicitly indicate the software or tools used in this manuscript’s empirical analysis, including the NLP and ML.

Response 3: Thank you for your valuable suggestion. In the manuscript, we have added the following statement: This experiment is conducted using Python 3.8.

Point 4: Improvement of the tables in the results section.

Comparing Tables A1 and A2, it is found that the ranking (in the Rank column) is not based on the average or total value of the four evaluation indicators (Accuracy, Specificity, Sensitivity, and AUC). It is suggested that the authors clearly state the ranking criteria for the models in this manuscript. In addition, the authors should add a column of data for the specific ranking indicators to the current tables, instead of simply listing the ranking results. This will help readers understand more intuitively the specific differences between each model.

For example, in Table A1, the average of the four evaluation indicators of the Logistic Regression model (0.6586) is higher than that of the Bagging model (0.6572), but the Rank of the Logistic Regression model is 6, whereas the Rank of the Bagging model is 2. It makes readers confused.

Response 4: Allow us to explain on this matter. In the first section of the empirical results analysis, titled Analysis of Empirical Results of Financial Crisis Early Warning Based on Traditional Financial Indicators, we have clarified the criteria used for ranking the models. Considering the length constraints of the manuscript, we emphasize that this ranking criterion applies to subsequent empirical results of the same nature. The specific statement is as follows:

This paper first conducts a test on the early warning effect of thirteen machine learning models using traditional financial indicators. The detailed results are shown in Table 3, reflecting the accuracy, specificity, sensitivity, and AUC of thirteen machine learning models. The last column of the table ranks the models based on the magnitude of their AUC values (the same below).

Point 5:

In the section of Analysis of Empirical Results of Financial Crisis Early Warning Based on Traditional Financial Indicators and MD&A Text Tone Indicator, there is a statement, “Although there is a slight increase in the mean value, there is still a decrease in some of the evaluation indicators measured by some models”. However, after comparing Tables A1 and A2, it is found that only two models (SVM and AdaBoost) have a decrease in evaluation indicators of specificity and AUC, while the rest have an increase. This is not a sufficient reason for the authors’ conclusion that the MD&A Text Tone Indicator is noisy. It is recommended that the authors strengthen their argument in this paragraph.

Response 5: Thank you for your valuable suggestion. Previously, we took into consideration that the empirical results indicate a very limited average improvement (less than 1%) in the early warning performance of models incorporating the MD&A text tone indicator compared to models using traditional financial indicators. Specifically, the differences in early warning performance between models, whether decreased or increased, are negligible. Therefore, we believed it brings noise. However, such an expression may not be sufficiently accurate. Unlike the noise introduced by text forward-looking indicator, the empirical results of incorporating the text tone Indicator do not provide further evidence of incremental information from the text tone indicator. Based on this, we have comprehensively revised the relevant statements in the manuscript, with the modifications tracked using tracked changes.

Response to Reviewer 4 Comments

Point 1: Is the manuscript technically sound, and do the data support the conclusions?

The manuscript presents an in-depth investigation into predicting financial crises in Chinese A-share listed companies, employing MD&A text readability, similarity, tone, and forward-looking indicators from 2011-2020. The authors claim that text readability and similarity are valuable for identifying financial crises, supporting existing research connecting these indicators to firm performance. However, they found that tone and forward-looking indicators did not enhance the identification of financial crises, as they can be manipulated by management. Utilizing machine learning models such as RF, Bagging, CatBoost, GBDT, XGBoost, and LightGBM, they provided a nuanced understanding of how combining text-linguistic features with traditional financial indicators can enhance early warning systems for financial crises. There are certain areas where the technical soundness could be improved, and where the data may not fully support the conclusions:

• Methodology Clarity: The methodology section could provide more detailed information about the data preprocessing, feature extraction, and the selection process for the machine learning models. Examples or specific rationale for the chosen methods would enhance the clarity.

• Control and Replication: The manuscript lacks adequate information about controls, replication strategies, and potential confounding factors, such as industry-specific patterns, economic conditions, etc.

• Sample Size Justification: A clear justification of the sample size would add to the rigor of the study and ensure that the findings are robust and generalizable.

Response 1: In response to Methodology Clarity, in addition to providing detailed explanations in the sections Evaluation Index System Construction, Sample Selection and Source of Data, and Data Processing and Descriptive Statistics in the paper, we have uploaded the code files and sample data to GitHub. The specific URL for accessing them is as follows. You can access them at the following URL: https://github.com/zzs0216/Financial-Crisis-Early-Warning-of-Chinese-Listed-Companies/tree/main (accessed on 19 August 2023).

In response to Control and Replication, allow us to provide further explanation. The paper does not study industry-specific patterns because its main contribution lies in measuring the enhancement effect of various text-linguistic feature indicators in MD&A texts on the early warning of financial crises, based on the same sample dataset. This study provides a clearer examination of the incremental information contained in different text-linguistic indicators, addressing the limitations of existing research. Furthermore, although industry-specific pattern analysis is valuable in the early warning of financial crises for listed companies, there are limitations when heavily relying on industry-specific analysis. These limitations encompass concerns regarding applicability, data availability, and model complexity. Specifically, (1) Applicability of general indicators: financial crises often exhibit a certain level of universality, extending beyond specific industries. Therefore, utilizing general indicators in financial crisis early warning can offer broader applicability and comparability. In contrast, excessive reliance on industry-specific analysis may restrict our ability to provide early warnings for other potential emerging industries. (2) Changes in industry specificity: industry-specific characteristics may undergo changes over time. A financial crisis early warning model for a particular industry may perform well, but its predictive ability could weaken with industry transformations or the emergence of new industries. Therefore, excessive reliance on industry-specific analysis may lead to neglect of industry changes or emerging industries. (3) Data availability: industry-specific patterns analysis relies heavily on a substantial amount of industry-specific data. However, for certain industries, the sample size of listed companies is small, which hinders the robustness of research conclusions. (4) Complexity of predictive models: introducing industry-specific patterns analysis increases the complexity of predictive models. This complexity can make the models difficult to comprehend and apply. In contrast, the use of general indicators is more straightforward and easier to understand, enabling their wider application in financial crisis early warning.

Indeed, based on our current study, future studies can further expand on industry-specific research, as well as the economic conditions you mentioned, such as economic booms, economic recessions, inflation, deflation, high unemployment rates, interest rate fluctuations, macroeconomic policies, market supply and demand dynamics, political risks, natural disasters, and various other economic conditions, to explore the early warning of financial crises in listed companies.

In response to Sample Size Justification, we have revised the section Sample Selection and Source of Data. The revised content is as follows:

Regarding financial crisis sample selection, scholars commonly consider a listed company being specially treated (ST) as a primary sign of being in a financial crisis [1-4]. Effective financial crisis warnings should have an early detection capability. Compared to the failure to meet debt obligations and bankruptcy, listed companies are often labeled as ST at an earlier stage. Thus, identifying ST status is more effective as an early warning. Therefore, this study selects companies in Shanghai and Shenzhen that firstly labeled as ST-listed from 2011 to 2020 as the sample for financial crises, totaling 481 companies over a 10-year period. The year in which a listed company experiences a financial crisis is defined as year T. ST refers to listed companies that have received special treatment for two consecutive years of losses, so the data in year T-2 can already show a financial crisis. To avoid overestimating the model’s prediction ability, this study utilizes data in year T-3 instead of year T-2, i.e., 2008 – 2017 data of both traditional financial indicators and text-linguistic feature. The selection of 2008 as the starting point is primarily driven by the implementation of new accounting standards for Chinese listed companies in 2007. This brought significant changes in the meaning and calculation methods of traditional financial indicators, resulting in lower comparability between data before and after 2007. Simultaneously, the global financial crisis erupted in 2008, leading to substantial changes in both the Chinese domestic and international economic environment. In response, China proactively provided new interpretations and adjustments to annual reports under the new accounting standards. This action enhanced the comparability of data after 2008. Additionally, the disclosure standards for MD&A texts were standardized after 2008, further contributing to the improved comparability of data after 2008. Therefore, based on these factors, we have chosen 2008 as the starting point for our research. 

Subsequently, this paper excludes samples of listed companies in finance and insurance, as well as those with missing data on traditional financial indicators in year T-3 and missing text similarity. This results in a final sample size of 416 ST-listed companies. Considering the unbalanced number of ST-listed companies and non-ST-listed companies, this paper will explain the treatment in section Data Processing and Descriptive Statistics. The total number of processed samples is 832, comprising 416 non-ST-listed companies and 416 ST-listed companies. The traditional financial indicators data are obtained from the China Economic and Financial Research Database, while the MD&A text is crawled from the Cninfo website and processed into MD&A text-linguistic feature indicators through natural language processing technology.

Point 2: Has the statistical analysis been performed appropriately and rigorously?

The manuscript employs various machine learning models, but the statistical analysis needs further clarification:

• Model Validation: More details on cross-validation, hyperparameter tuning, and model evaluation metrics are needed to ensure the validity of the results.

• Comparison of Models: A thorough statistical comparison of the selected models against others would clarify why these specific models were chosen and how they outperform others.

Response 2: (1) About the detailed operation of cross-validation, this paper has a specific description, the content is as follows: 

To enhance the generalization ability of the machine learning model and prevent overfitting, a 10-fold cross-validation method is employed in this study. This method divides the sample dataset into 10 copies, and takes turns to using 9 of them as the training set and 1 as the test set. The training set is used to train the machine learning model, while the test set evaluates its performance. To further increase the reliability and reduce result variability, a 10 times 10-fold cross-validation is conducted, and the averaged results serve as the estimation of the model’s effect.

(2) We have uploaded the relevant code for hyperparameter selection to GitHub, providing a clear understanding of the specific details of the hyperparameters. The specific URL for accessing the code is as follows: https://github.com/zzs0216/Financial-Crisis-Early-Warning-of-Chinese-Listed-Companies/tree/main (accessed on 19 August 2023).

(3) In the section Machine Learning Model Early Warning Effect Evaluation Indicator, we introduce the model evaluation metrics: accuracy, sensitivity, specificity, and AUC value. We further explain the significance of these indicators and provide their respective calculation formulas. In the subsequent empirical studies, the results of each sub-study displayed these four indicators, along with comprehensive explanations.

In response to Comparison of Models, please allow us to provide the following explanation. In the field of financial crisis prediction methods for listed companies, relevant research has evolved from univariate analysis to multivariate analysis and, more recently, the widespread adoption of machine learning methods. Therefore, this study investigates the predictive effectiveness of 13 mainstream machine learning models, which encompass traditional machine learning models, tree-based machine learning models, and ensemble machine learning models. It can be considered a comprehensive exploration rather than a specific one. We discuss the characteristics of these models in the section Machine Learning Models.

Point 3: Have the authors made all data underlying the findings in their manuscript fully available?

The manuscript does not provide complete information regarding the availability of the underlying data:

• Data Availability Statement: The Data Availability Statement should explicitly mention where the data can be accessed and whether there are any restrictions on data sharing.

• Data Description: A more thorough description of the dataset, including the web crawler technology used, and specific details about the extracted features, would enhance the reproducibility of the study.

Response 3: Thank you for your valuable suggestion. We have uploaded the data to the shared repository and it is described in Data availability and access. The description is as follows: the data and code of this study can be downloaded by visiting https://github.com/zzs0216/Financial-Crisis-Early-Warning-of-Chinese-Listed-Companies/tree/main (accessed on 19 August 2023).

In addition, we added descriptive statistics in the chapter "Data Processing and Descriptive Statistics", and the descriptive statistics of traditional financial indicators and text-linguistic feature indicators are shown in Table 2.

Table 2. Descriptive statistics of traditional financial indicators and text-linguistic feature indicators

Variable Obs Mean Std. Dev. Min Max

Earnings per share 832 0.14 0.57 -5.88 4.49 

Operating profit ratio 832 -0.21 4.18 -116.76 13.97 

Debt assets ratio 832 0.56 0.43 0.02 8.00 

Current ratio 832 1.45 1.57 0.00 21.12 

Cash coverage ratio 832 0.09 0.52 -9.22 4.31 

Interest cover 832 48.67 1,244.86 -1,471.08 35,725.11 

Accounts receivable turnover ratio 832 329.61 4,671.90 0.27 124,214.10 

Sales-to-cash ratio 832 1.02 0.90 0.12 25.15 

Operating cash per share 832 0.22 0.85 -6.65 9.42 

Net profit growth rate 832 -131.71 1,301.70 -23,572.69 13,737.70 

Tone Indicator 832 0.40 0.17 -0.22 1.00 

Forward-Looking Indicator 832 0.01 0.00 0.00 0.02 

Readability Indicator 832 -21.88 4.26 -42.82 -11.20 

Similarity Indicator 832 0.60 0.16 0.03 0.97 

ST (Not St is 1, St is 2) 832 1.50 0.50 1 2

Regarding web crawler technology, we consider that this article is not focused on technical aspects. Therefore, the discussion of web crawler technology, which is not original, is limited to its role in obtaining text data without further elaboration. In addition, considering that the existing manuscript already has a large amount of content, adding a detailed introduction to web crawler technology may result in fragmented content and a lack of focus. Hence, we kindly request your approval and understanding in this matter.

Finally, the calculation methods for the ten traditional financial indicators and the four text-linguistic feature indicators are explained in the section Traditional Financial Indicators and the section MD&A Text-Linguistic Feature Indicators, respectively. The relevant data has also been uploaded to the public repository.

Point 4: Is the manuscript presented in an intelligible fashion and written in standard English?

The manuscript is mostly well-written but suffers from some grammatical errors and unclear expressions:

• Language and Grammar: Some sentences are complex and could be broken down into simpler structures to improve readability. A thorough proofreading is recommended.

• Figure and Table Descriptions: Additional explanations for figures and tables, along with concise captions, would enhance understanding.

Response 4: Thank you for your valuable suggestion. We have simplified complex sentences to improve the readability. We have also reviewed the manuscript for grammar issues and made necessary corrections. Moreover, we have revised the table captions and improved accompanying descriptions. The modifications have been tracked using tracked changes in the revised manuscript.

Point 5: Review Comments to the Author

This study presents a novel approach to early warning of financial crises in listed companies, leveraging text analysis and machine learning. While the concept is promising, there are several areas where the manuscript could be strengthened:

• Enhance the Technical Details: More information on methodology, statistical analysis, and data availability is essential to increase the rigor and reproducibility of the study.

• Consider Industry-Specific Analysis: Incorporating industry-specific characteristics might provide a more nuanced understanding and enhance the model's predictive power.

• Improve Language and Presentation: Proofreading for grammatical errors and enhancing figure and table descriptions will improve the manuscript's readability.

The addressed areas of improvement will not only enhance the technical soundness of the paper but also contribute to a broader understanding and applicability of the proposed methods. This research has the potential to make an impact, and I wish you all the best with its continued development.

Response 5: Regarding Enhance the Technical Details, in addition to providing explanations in the paper, we have uploaded the code files and dataset to a public repository to facilitate reproducibility and enable others to study and utilize them. As for Consider Industry-Specific Analysis, it can serve as a potential avenue for future research, and the specific reasons and explanations have been described in Response 1. Lastly, in response to Improve Language and Presentation, we have revised and improved the language and presentation.

---

## [Decision Letter · Decision Letter 2]

7 Sep 2023

Financial Crisis Early Warning of Chinese Listed Companies Based on MD&A Text-linguistic Feature Indicators

PONE-D-23-13667R2

Dear Dr. Huayong Niu,

We’re pleased to inform you that your manuscript has been judged scientifically suitable for publication and will be formally accepted for publication once it meets all outstanding technical requirements.

Kind regards,

Normaizatul Akma Saidi, Ph.D

Academic Editor

PLOS ONE

Additional Editor Comments (optional):

Reviewers' comments:

Reviewer's Responses to Questions

**Comments to the Author**

1. If the authors have adequately addressed your comments raised in a previous round of review and you feel that this manuscript is now acceptable for publication, you may indicate that here to bypass the “Comments to the Author” section, enter your conflict of interest statement in the “Confidential to Editor” section, and submit your "Accept" recommendation.

Reviewer #1: All comments have been addressed

Reviewer #3: All comments have been addressed

2. Is the manuscript technically sound, and do the data support the conclusions?

Reviewer #1: Yes

Reviewer #3: Yes

3. Has the statistical analysis been performed appropriately and rigorously? 

Reviewer #1: Yes

Reviewer #3: Yes

4. Have the authors made all data underlying the findings in their manuscript fully available?

Reviewer #1: Yes

Reviewer #3: Yes

5. Is the manuscript presented in an intelligible fashion and written in standard English?

Reviewer #1: Yes

Reviewer #3: Yes

6. Review Comments to the Author

Reviewer #1: The authors have adequately addressed my comments raised in a previous round of review and I feel that this manuscript is now acceptable for publication.

Reviewer #3: The authors have addressed my comments raised from the previous round of review. I suggest that this manuscript be accepted.

7. PLOS authors have the option to publish the peer review history of their article (what does this mean?). If published, this will include your full peer review and any attached files.

Reviewer #1: No

Reviewer #3: No

---

## [Editor Report · Acceptance letter]

13 Sep 2023

PONE-D-23-13667R2 

Financial Crisis Early Warning of Chinese Listed Companies Based on MD&A Text-linguistic Feature Indicators 

Dear Dr. Niu:

I'm pleased to inform you that your manuscript has been deemed suitable for publication in PLOS ONE. Congratulations! Your manuscript is now with our production department. 

Kind regards, 

on behalf of

Dr. Normaizatul Akma Saidi 

Academic Editor

PLOS ONE